# CoreMatching: A Co-adaptive Sparse Inference Framework with Token and Neuron Pruning for Comprehensive Acceleration of Vision-Language Models

Qinsi Wang[1]   Hancheng Ye[1]   Ming-Yu Chung[1]   Yudong Liu[1]   Yueqian Lin[1]   Martin Kuo[1]   Mingyuan Ma[1]
Jianyi Zhang[1]   Yiran Chen[1]

## Abstract

Vision-Language Models (VLMs) excel across diverse tasks but suffer from high inference costs in time and memory. Token sparsity mitigates inefficiencies in token usage, while neuron sparsity reduces high-dimensional computations, both offering promising solutions to enhance efficiency. Recently, these two sparsity paradigms have evolved largely in parallel, fostering the prevailing assumption that they function independently. However, a fundamental yet underexplored question remains: Do they truly operate in isolation, or is there a deeper underlying interplay that has yet to be uncovered? In this paper, we conduct the first comprehensive investigation into this question. By introducing and analyzing the matching mechanism between **Core Neurons** and **Core Tokens**, we found that key neurons and tokens for inference mutually influence and reinforce each other. Building on this insight, we propose **CoreMatching**, a co-adaptive sparse inference framework, which leverages the synergy between token and neuron sparsity to enhance inference efficiency. Through theoretical analysis and efficiency evaluations, we demonstrate that the proposed method surpasses state-of-the-art baselines on ten image understanding tasks and three hardware devices. Notably, on the NVIDIA Titan Xp, it achieved $5\times$ FLOPs reduction and a $10\times$ overall speedup. Code is released at https://github.com/wangqinsi1/2025-ICML-CoreMatching/tree/main.

[1]Department of Electrical and Computer Engineering, Duke University, North Carolina, USA. Correspondence to: Qinsi Wang <qinsi.wang@duke.edu>, Jianyi Zhang <jianyi.zhang@duke.edu>.

*Proceedings of the 42nd International Conference on Machine Learning*, Vancouver, Canada. PMLR 267, 2025. Copyright 2025 by the author(s).

## 1. Introduction

Large language models (LLMs) have achieved outstanding performance in various applications and have exerted a significant influence on our daily life (Brown, 2020; Chowdhery et al., 2023; Touvron et al., 2023a; Kuo et al., 2025; Qinsi et al.). As LLMs become increasingly aware of the physical world, researchers have discovered their potential for understanding visual information (Lin et al., 2023; Liu et al., 2024a; 2025b). As a result, a series of vision-language models (VLMs) such as LLaVA (Liu et al., 2024b), Blip(Li et al., 2022), and Llama(Touvron et al., 2023b) have been introduced, demonstrating impressive performance on tasks like image-based question answering. However, because of the requirement of long image-token inputs, VLMs typically demand more time and memory for inference than LLMs, limiting their practical adoption in real-world scenarios.

**Token Sparsity**, which exploits the high degree of redundancy among image tokens, is a promising solution to this challenge(Ye et al., 2024; Huang et al., 2024). Researchers have found that VLMs can retain strong performance using as little as 10% of the total tokens. This has led to extensive interest in determining which tokens are essential. For example, PruMerge (Shang et al., 2024) uses the average attention scores between image tokens and text tokens to measure token importance and retains 20% of the tokens with only minor performance loss; FastV (Chen et al., 2025) uses the total attention scores received by other token and finds that more than half of the tokens can be discarded from the second layer. Nevertheless, it is worth noting that most existing methods rely on attention scores as a guide for selecting important tokens, but their validity and accuracy have not been thoroughly examined.

Another effective approach to accelerating inference is model-internal sparsity, particularly adaptive **Neuron Sparsity**. It leverages the highly sparse activations in feedforward network (FFN) layers to skip the computation of inactive neurons, thereby reducing the computational in LLMs. For instance, methods such as Dejavu (Liu et al., 2023) and PowerInfer (Song et al., 2023) employ MLP-based predictors to identify which neurons are activated for a given input, achieving up to 93% prediction accuracy. Fur-

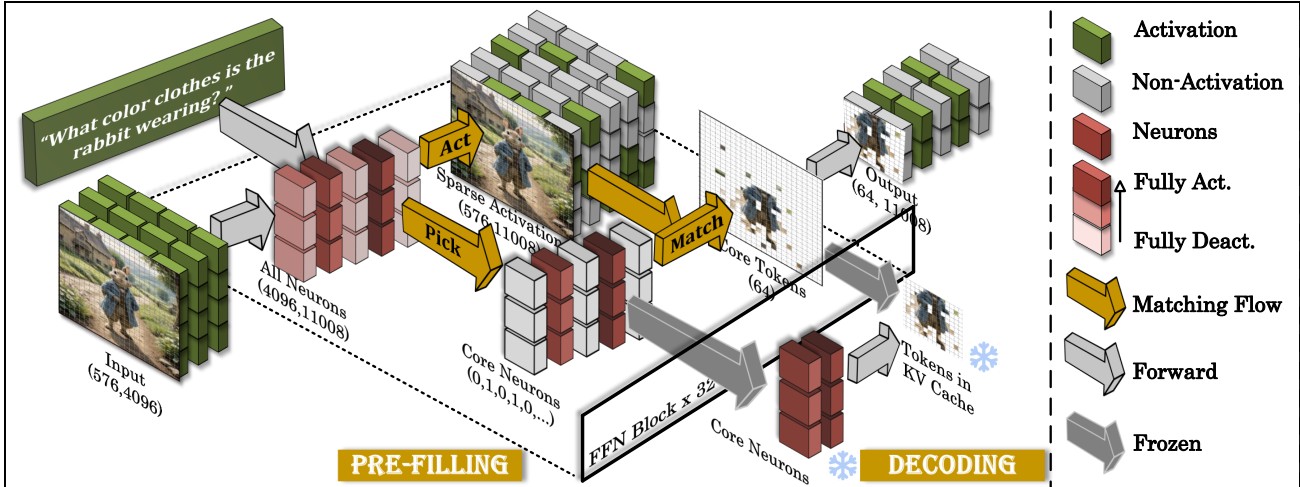

*Figure 1.* Schematic diagram of CoreMatching. In the Pre-filling stage, CoreMatching calculates Core Neurons in the FFN block based on the activation. Core Neurons are the most frequently activated group of neurons. Afterwards, CoreMatching matches the neurons activated by different tokens with the core neurons, and selects a group of tokens with the largest intersection as the Core Tokens. Only the Core Tokens are passed to the subsequent layers. During the decoding stage, the model only uses Core Neurons for calculations, and there are only core tokens in the kv cache. CoreMatching achieves comprehensive acceleration for inference of VLMs.

ther, CoreInfer (Wang et al., 2024) defines core neurons as the subset of neurons most frequently activated by an input sentence, demonstrating that only these core neurons are needed to maintain performance. While neuron sparsity has shown great promise in LLMs, it remains underexplored and underutilized in VLMs.

Although both token sparsity and neuron sparsity can individually accelerate the model, each has limitations in practical applications. Token sparsity primarily speeds up the pre-filling stage and can only provide limited acceleration during decoding by reducing key-value operations. In contrast, neuron sparsity primarily accelerates the decoding stage but cannot achieve a high speedup ratio in the pre-filling stage due to the large number of tokens and the resulting low level of sparsity. Hence, combining these two forms of sparsity is a promising approach to achieving comprehensive acceleration. However, an interesting question that has been overlooked in previous work is: **What is the relationship between these two sparse spaces?**

To answer this question, we first experimentally verify the existence of neuron sparsity in VLMs and the effectiveness of core neurons, which are the most important neurons determined by the activation distribution of all input tokens. Furthermore, we investigate the alignment between core neurons and tokens. By analyzing how core neurons influence token inference, we uncover an insightful matching pattern: **tokens whose activated neurons most closely match the core neurons correspond to the most critical part for determining the output.** Inspired by this, we define core tokens as the set of tokens that have the largest intersection of activated neurons with the core neurons among all tokens.

Building on these insights, we propose CoreMatching, a

co-adaptive inference framework. As illustrated in Fig. 1, CoreMatching requires a single step in the pre-filling stage to jointly compute Core Neurons and Core Tokens, thereby achieving sparsity along both the token and neuron dimensions. Furthermore, we conduct a detailed analysis of the theoretical and practical benefits of CoreMatching. Theoretically, we propose a projection-guided criterion for evaluating the importance of tokens, which takes into account not only attention scores but also angular information. We analyze the effectiveness and efficiency of this criterion and theoretically analyze the proportional relationship between core tokens and the criterion. Empirically, we implement and evaluate CoreMatching across various tasks and hardware, showing that it exceeds the performance of state-of-the-art token sparsity approaches. CoreMatching delivers comprehensive inference acceleration due to its multi-dimensional sparsity, achieving a 2.1× speedup in the pre-filling stage and a 9.2× speedup in the decoding stage. In summary, our contributions are as follows:

- We introduce Core Tokens, which are those tokens that activate the largest number of core neurons. Experiments show that core tokens consistently capture the subset of tokens most relevant to the output.

- We present a Projection-guided Criterion and leverage it to explain why Core Tokens outperform those selected by previous attention score-based methods.

- We propose CoreMatching, a co-adaptive inference framework. Experimental results confirm that it not only surpasses baselines on the performance of various tasks but also achieves comprehensive acceleration.

*Table 1.* Verification of the importance of core neurons. We retain different proportions of core neurons on Llava-1.5-7b, and we calculate the accuracy of the model on TextVQA.

| Keep Ratios | 0.2 | 0.4 | 0.6 | 0.8 | 1.0 |
|---|---|---|---|---|---|
| Accuracy | 45.1% | 53.2% | 55.8% | 56.3% | 57.8% |

*Figure 2.* Verification of the predictability of core neurons. We visualized the core neurons of the 25-th layer of Llava-1.5-7b when input text token at different lengths. $\rho = 0.2, \beta = 0.4$. We selected the first 256 neurons. It can be seen that when the input semantics are sufficient, core neurons are almost unchanged.

## 2. Intrinsic Relations of Two Paradigms

In this section, to delve deeper into the relationship between two sparse spaces, we propose a novel interaction paradigm between tokens and neurons. Through this approach, we investigate how important neurons and tokens mutually influence and determine each other.

### 2.1. Identifying Core Neurons for Tokens

In the FFN block in LLMs, there are typically two linear layers, $W_u \in \mathbb{R}^{N_1 \times N_2}, W_d \in \mathbb{R}^{N_2 \times N_1}$. For a single token, denote the input representation of the FFN layer as $x$, the output $y$ can be expressed as:

$$A = \sigma(xW_u), \quad y = AW_d \quad (1)$$

where $\sigma$ represents the activation function, such as ReLU or SiLU. Intermediate output $A = \{a_n\}_{n=1}^{N_2}$, where $a_n$ is the activation value of the $n$-th neuron in $W_u$.

**Core Neurons.** The concept of Core Neurons was first introduced in CoreInfer (Wang et al., 2024), which represents a group of neurons that are most important for input to maintenance performance.

Specifically, for a single token $x$, the token-wise core neurons $\mathcal{C}_\rho(x)$ are defined as the neurons with the top $\rho$ maximum positive activation values, that is,

$$\mathcal{C}_\rho(x) = \{n \mid a_n \geq \text{Percentile}(A^+, \rho)\}, \quad (2)$$

where $A^+ = \{a_n \mid a_n > 0, a_n \in A\}$ represents the set of positively-activated neurons, and $\text{Percentile}(A^+, \rho)$ denotes the $\rho$-th percentile of the positive activation.

For a sentence $\boldsymbol{s} = [x_1, x_2, \ldots, x_M]$ containing $M$ tokens. The sentence-wise core neurons $\mathcal{C}_\rho^\beta(\boldsymbol{s})$ are defined as the top $\beta$ of neurons that appear most frequently in the core neurons of all tokens, $\bigcup_{i=1}^{M} \mathcal{C}_\rho(x_i)$, which is formulated as

$$\mathcal{C}_\rho^\beta(\boldsymbol{s}) = \{n \mid f_\rho(n; \boldsymbol{s}) \geq \text{Percentile}(f_\rho(\boldsymbol{s}), \beta)\}, \quad (3)$$

where $f_\rho(\boldsymbol{s})$ denotes the count set of each neuron marked as the core neuron across all tokens, as defined in Eq. (4).

$$f_\rho(\boldsymbol{s}) = \{f_\rho(n; \boldsymbol{s})\}_n = \{\sum_{m=1}^{M} \mathbb{I}(n \in \mathcal{C}_\rho(x_m))\}_n, \quad (4)$$

where $\mathbb{I}(\cdot)$ is an indicator function that returns `one` if $n$ is in $\mathcal{C}_\rho(x_m)$ else `zero`. Percentile$(f_\rho(\boldsymbol{s}), \beta)$ denotes the $\beta$-th percentile of $f_\rho(\boldsymbol{s})$.

**Effectiveness.** For an input $s$, core neurons are the subset of neurons that are most frequently activated and exhibit the highest activation values. In LLMs, core neurons have two key characteristics. First, they are crucial for the inference, as only core neurons retained can maintain the performance. Second, they exhibit predictability; when input is sufficiently long and semantically rich, the core neurons identified during the pre-filling stage closely align with those in the decoding stage.

To validate that core neurons applicable to VLMs, we conducted experiments on Llava–1.5-7b (Liu et al., 2024a). As shown in Tab. 1 and Fig. 2, core neurons also exhibit these two characteristics on Llava. Specifically, by retaining only 40% of the neurons, the performance decreases by merely 3%. This demonstrates that in VLMs, given a specific input, we can accelerate inference by preserving only a small set of core neurons. Note that the calculation of core neurons takes into account the activation distribution of all tokens. Therefore, **core neurons are essentially a part of neurons that capture the most critical features, which are determined by all tokens input.**

### 2.2. Deriving Core Tokens from Core Neurons

In the previous section, we defined core neurons, which are selected by counting the key activations determined by all input tokens. In this section, we explore how core neurons in turn select important tokens.

**Matching of Neurons and Tokens.** To explore the relationship between core neurons and tokens. We first study the impact of core neurons on different tokens. Specifically, we explore the changes in token inference when only core neruons are retained compared to using all neurons.

For a single token $x$, the set of neurons activated by it can be denoted as $\Gamma(x) = \{n \mid a_n(x) > 0\}$. When only the core neurons are retained, token $x$ can only activate neurons in the intersection $\Gamma(x) \cap \mathcal{C}_\rho^\beta(s)$. Consequently, information that token $x$ is able to transmit, i.e., $I(x)$, can be regarded as proportional to the number of intersections between the neurons it activates and core neurons,

$$I(x) \propto \left| \Gamma(x) \cap \mathcal{C}_\rho^\beta(s) \right|. \quad (5)$$

Therefore, to analyze how much information core neurons retain for different tokens, we can equivalently measure the number of core neurons each token activates.

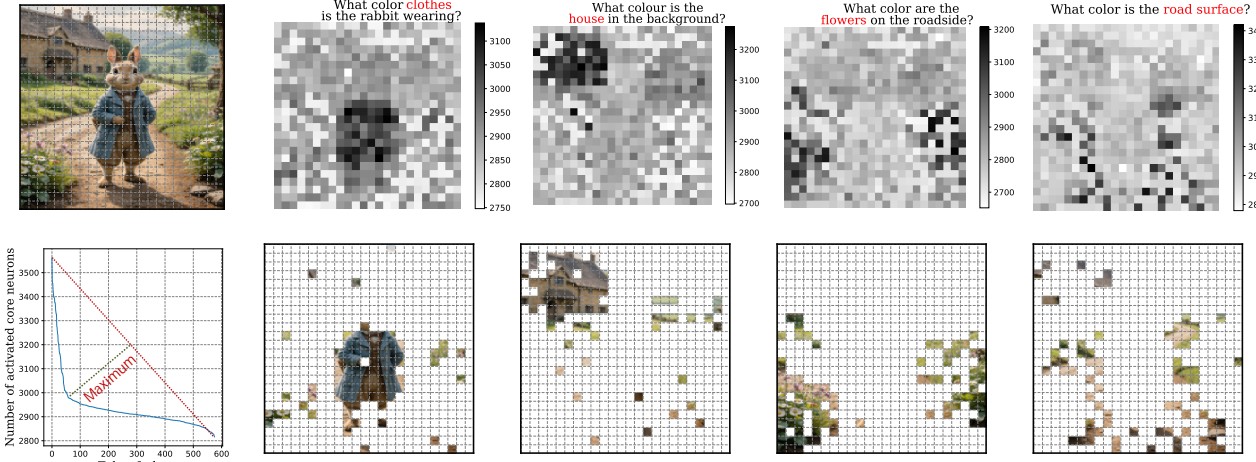

*Figure 3.* (Upper) Distribution of $\left|\Gamma(x) \cap \mathcal{C}_\rho^\beta(s)\right|$ of image token. The experiment was conducted on Llava-1.5-7b, and we selected the 10th layer. The input image is the rabbit on the left, and the input text is the text token above the image. We use red font to emphasize the key points of the text token. (Note that since core neurons themselves account for 40% of neurons, intersection of about 2000 can be regarded as random sample.) (Lower) Core token under different inputs. The left is the schematic diagram of the maximum geometric distance method to select the threshold. The right side is the core token retained under the distribution of the corresponding image above.

We conduct experiments on LLaVA-1.5-7B, focusing on the number of core neurons activated in the 10th layer for various tokens across different text inputs. The results, illustrated in Fig. 3 (Upper), reveal intriguing patterns. Specifically, image tokens that exhibit stronger correlations with the text prompt significantly activate more core neurons than other image tokens. Notably, even for the same image, the distribution of activated core neurons varies depending on the emphasis of the text prompt. Image tokens most relevant to the text consistently trigger a substantially higher activation of core neurons.

From the above analysis, we know that core neurons can preserve the information flow of important tokens while suppressing that of unimportant tokens. **The subset of tokens that activate the largest number of core neurons are the most critical for the inference.**

**Core Tokens.** Motivated by the fact that core neurons can guide the selection of important tokens, we define core tokens here and verify their effectiveness.

From the previous analysis, tokens that exhibit a higher correlation with text tokens activate more core neurons. Therefore, for input $s$, we can define core tokens $\mathcal{T}_\rho^\beta(s)$ as tokens activate the largest number of core neurons,

$$\mathcal{T}_\rho^\beta(s) = \left\{ x_m \,\middle|\, \left|\Gamma(x_m) \cap \mathcal{C}_\rho^\beta(s)\right| \geq T_k \right\}, \quad (6)$$

where $T_k$ is the "knee" threshold derived from the distribution of $\left|\Gamma(x) \cap \mathcal{C}_\rho^\beta(s)\right|$ for all tokens in $s$ using the maximum geometric distance method (Garg & Walker, 1990). Fig. 3 illustrates how $T_k$ is obtained. Notably, since $T_k$ is computed adaptively based on the data distribution, there is no need to manually fix the number of core tokens.

Fig. 3 (Lower) shows the retained core tokens under different text inputs. It can be observed that, for various text tokens, core tokens are consistently the most relevant to the text tokens. Moreover, the core tokens adaptively retain the most critical portion of the tokens based on the input, without being constrained by a fixed number limit.

In Section 4, we empirically demonstrate that core tokens represent the most crucial set of tokens for maintaining performance. By retaining only these core tokens during inference, the model can achieve nearly lossless performance.

## 3. CoreMatching

In this section, building on our previous findings, we propose CoreMatching, a co-adaptive inference framework. We further provide theoretical analysis explaining why core neurons can guide core tokens.

### 3.1. Overall Framework

As shown in Fig. 1, CoreMatching precomputes core neurons at each FFN layer during pre-filling, and identifies core tokens at layer $l$ and discards non-core tokens. During decoding, the model predicts the next token using the prefetched core neurons and core tokens. CoreMatching pruned unimportant tokens to speed up the pre-filling stage. Meanwhile, the reduced number of neurons also accelerates the decoding stage. Our method is training-free, and plug-and-play. Algorithm is provided in the Appendix .B.

### 3.2. Theoretical Advantage

To analyze why core tokens are effective, we first explore the question: *How can we evaluate the importance of image tokens in VLM?* In existing research (Shang et al., 2024; Chen et al., 2025), the attention scores between image tokens and text tokens are widely used as a metric for evaluation. However, no work has yet analyzed its optimality. Therefore, in this section, we first propose a superior criterion to quantify the importance of image tokens. Furthermore, we

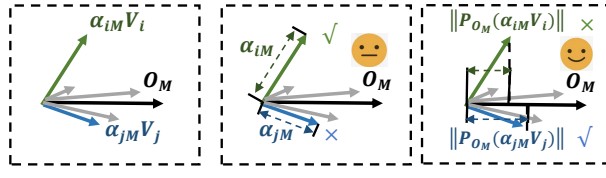

(a) Forms of $O_M$    (b) Attention Score    (c) Projection Value

*Figure 4.* Diagram of attention score and projection value. ✓ indicates the token is reserved under this matric. ✗ indicates discarded.

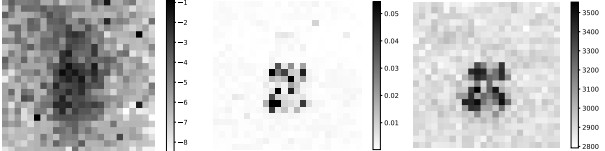

(a) Attention score    (b) Projection Value    (c) Core Tokens

*Figure 5.* Comparison of three metrics. The input is the rabbit in Fig. 3 and "What color clothes is the rabbit wearing?". Experiment is conducted on Llava-1.5-7b and the 10-th layer is selected.

mathematically derive and demonstrate that core tokens are proportional to this criterion.

**Projection-guided Criterion.** In the inference of VLM, the only module where tokens influence each other is the attention mechanism. And since the final output is determined by the last token in the input, we only need to focus on how all image tokens affect this last token.

For a single token, suppose its input to the Attention block is $y$, where $y = \sigma(x W_u) W_d$. Consider a sequence of tokens $[y_1, y_2, \ldots, y_M]$, for the last token $y_M$, its computation in the Attention layer can be expressed as:

$$\hat{y}_M = \text{LayerNorm}(y_M), \quad V_i = \hat{y}_i W_v,$$
$$\alpha_{iM} = Softmax\big((\hat{y}_i W_q)(\hat{y}_M W_k)^T\big), \quad (7)$$
$$O_M = \alpha_{1M} V_1 + \alpha_{2M} V_2 + \cdots + \alpha_{MM} V_M,$$

where $\alpha_{iM}$ is the attention score between the $i$-th and the $M$-th token. $O_M$ is the output vector of the $M$-th token.

As shown in Fig. 4 (a), when we focus on the $O_M$, we can observe that $O_M$ is the sum of multiple vectors. This indicates that $O_M$ depends not only on the attention scores $\alpha_{iM}$, but also on the angle between $V_i$ and $O_M$. Based on this insight, we propose that the Projection-guided Criterion for evaluating the influence of the $i$-th token on the $M$-th token should be the projection of $\alpha_{iM} V_i$ onto $O_M$, given by

$$\big\|\text{Proj}_{O_M}(\alpha_{iM} V_i)\big\| = \|\alpha_{iM} V_i\| \cos\big(\angle(V_i, O_M)\big). \quad (8)$$

where $\big\|\text{Proj}_{O_M}(\alpha_{iM} V_i)\big\|$ is the projection value of $\alpha_{iM} V_i$ on $O_M$. And $\cos\big(\angle(V_i, O_M)\big)$ is the cosine value of the angle between the $V_i$ and $O_M$ vectors. Fig. 4 provides a schematic illustration of this projection-based metric. Furthermore, in Fig. 5 (a) and (b), we compare the results of using the attention score as an evaluation metric with those



(a)$W_q W_k$    (b) $W_d W_d^T$    (c)Attention Score

*Figure 6.* (a) (b) Numerical visualization of $W_q W_k$ and $W_d W_d^T$. They are approximately $I$. (c) Visualization of $\alpha$. It can be seen that the $\alpha_{ii}$ (diagonally) is much higher than the others.

obtained using the projection magnitude. Obviously, the projection magnitude filters out important tokens more effectively, whereas the attention score alone considers only the vector magnitude and ignores angular information, which introduces more noise.

**Theoretical Analysis.** Although the projection value is a superior evaluation metric, its computation is both complex and time-consuming. Therefore, we show that it is actually related to the proposed core token.

To demonstrate this, we first introduce two observations:

*Observation 1.* $W_q$ and $W_k$ are nearly approximately orthogonal to each other, i.e., $W_q W_k^T \approx \theta I$. $\theta$ is a constant. $(W_o, W_u, W_d)$ are approximately orthogonal matrices, i.e., $W W^T \approx \lambda I$. $\lambda$ is a constant.

We present experimental evidence supporting this observation in Fig. 6, and provide comprehensive experiment and more detailed discussion in Appendix C. This is a fascinating phenomenon in VLMs. In fact, further investigation reveals that such matrix orthogonality was previously proven to exist and be effective in MLPs and CNNs. For example, (Li et al., 2019; Bansal et al., 2018) have shown that this orthogonality helps improve the training stability and generalization ability of the model.

*Observation 2. For activation function output vectors $A_i$ and $A_M$, $cos(\angle(A_i, A_M))$ is proportional to the number of intersections of activated neurons, i.e., $\big|\Gamma(x_i) \cap \Gamma(x_M)\big|$.*

This observation is illustrated in Fig. 7. An intuitive understanding is that if $x_i$ and $x_M$ activate more of the same neurons, $A_i$ and $A_M$ will have more positive values in common positions, making $\cos(A_i, A_M)$ larger. We provide more visualizations in Appendix C.

Building on these two observations, we provide insights into how projection values are influenced in LLMs. Following the inference order, we first analyze the impact of input vector $y$ on projection values, then examine how the activation layer affects these input vectors.

*Theoretical Insight 1. The projection values between the $i$-th and $M$-th token, i.e., $\big\|\text{Proj}_{O_M}(\alpha_{iM} V_i)\big\|$, is proportional to the cosine value of the angle between $y_i$ and $y_M$.*

*Justification.* To analyze how $y_i$ and $y_M$ influence the pro-

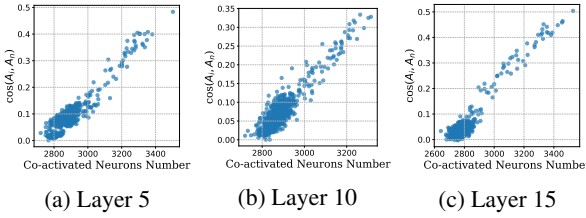

(a) Layer 5    (b) Layer 10    (c) Layer 15

*Figure 7.* The distribution of $cos(A_i, A_M)$ and $\left|\Gamma(x_i) \cap \Gamma(x_M)\right|$ of LLaVA-1.5-7b, clearly shows a proportional relationship.

jection value, we first decompose it into a form related to these two tokens. From Eq. 7, $O_M$ is a sum of vectors in different directions. Since the self-attention score $\alpha_{MM}$ is typically much higher than $\alpha_{iM}$ for other tokens (as shown in Fig. 6 (c)), we can simplify the projection by assuming that $O_M$ is primarily determined by $\alpha_{MM}V_M$, i.e.,

$$\left\|\text{Proj}_{O_M}(\alpha_{iM}V_i)\right\| \approx \left\|\alpha_{iM}V_i\right\| \cos(\angle(V_i, V_M)), \quad (9)$$

where $\alpha_{iM} = Softmax\left((\hat{y}_i W_q)\left(\hat{y}_M W_k\right)^T\right)$. Since the Softmax function is monotonic, that is, $Softmax(x) \propto x$. We can have $\alpha_{iM} \propto (\hat{y}_i W_q)\left(\hat{y}_M W_k\right)^T$. Therefore,

$$
\begin{aligned}
&\left\|\alpha_{iM}V_i\right\| \cos(\angle(V_i, V_M)) \\
&\propto (\hat{y}_i W_q)\left(\hat{y}_M W_k\right)^T \|V_i\| \cos(\angle(V_i, V_M)) \\
&= \langle \hat{y}_i W_q, \hat{y}_M W_k \rangle \langle V_i, V_M \rangle / \|V_M\| \\
&= \left(\hat{y}_i(W_q W_k^T)\hat{y}_M^T\right)\left(\hat{y}_i(W_v W_v^T)\hat{y}_M^T\right) / \|V_M\|
\end{aligned}
\quad (10)
$$

Based on the Observation 1, $W_q W_k^T \approx \theta I$, $W_v W_v^T \approx \lambda I$. Therefore, combining Eq. 9 and Eq. 10, we can have

$$\left\|\text{Proj}_{O_M}(\alpha_{iM}V_i)\right\| \propto \theta\lambda\langle \hat{y}_i, \hat{y}_M \rangle\langle \hat{y}_i, \hat{y}_M \rangle / \|V_M\| \quad (11)$$

Given that $\hat{y}$ is the result of $y$ after LayerNorm, $\hat{y}$ and $y$ share the same direction, and $\|\hat{y}\| = 1$, it holds that $\langle \hat{y}_i, \hat{y}_M \rangle = cos(\angle(y_i, y_M))$. We can have

$$\left\|\text{Proj}_{O_M}(\alpha_{iM}V_i)\right\| \propto \cos(\angle(y_i, y_M)), \quad (12)$$

which is consistent with Insight 1.

Eq. 12 shows that the angle between different tokens directly affects their mutual interaction in attention layer. The closer the angles of two tokens, the greater their mutual influence. **Therefore, the angle should be an essential indicator for token importance evaluation, which has not been fully demonstrated and utilized in previous work.**

Furthermore, we analyze how the activation influences the relative angle between $y_i$ and $y_M$.

*Theoretical Insight 2.* $cos(\angle(y_i, y_M))$ *is proportional to the number of intersections of neurons activated in the previous activation layer, i.e.,* $\left|\Gamma(x_i) \cap \Gamma(x_M)\right|$.

*Justification.* To analyze how the activation layer affects $cos(\angle(y_i, y_M))$, we first decompose its computation formula. Suppose the activation output of the $i$-th token is

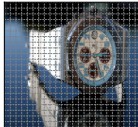 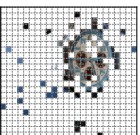 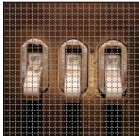 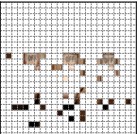

(a) What time is it now?    (b) What's the word on the button?

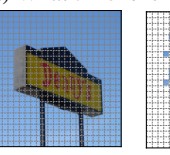 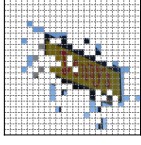 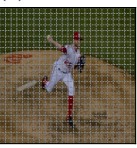 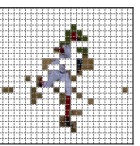

(c) What's the word on the sign?(d) What's the person wearing?

*Figure 8.* Examples of CoreMatching sampled tokens.

$A_i$, and $y_i = A_i W_d$. Based on Observation 1, $W_d$ is a scalar multiple of a unitary self-orthogonal matrix, applying the same rotation to any input while preserving the inner product and angle between any two input vectors (proof is provided in Appendix C). Thus, we can have:

$$\cos(\angle(y_i, y_M)) = \cos(\angle(A_i, A_M)) \quad (13)$$

Furthermore, based on Observation 2, $cos(\angle(A_i, A_M)) \propto \left|\Gamma(x_i) \cap \Gamma(x_M)\right|$. Combined with Eq. 19, we can get

$$\cos(\angle(y_i, y_M)) \propto \left|\Gamma(x_i) \cap \Gamma(x_M)\right|. \quad (14)$$

which is consistent with Insight 2.

**Eq. 14 shows that activation layers adjust token angles by controlling the intersections of their activated neurons.** More shared activated neurons lead to smaller angles and greater mutual influence.

**Correlation with Core Tokens.** Based on the predictability of core neurons, when the semantics of the input is stable enough, the neurons activated by $x_M$ are approximately the same as those of the core neurons, $\Gamma(x_M) \approx \mathcal{C}_\rho^\beta(s)$. Therefore, by combining Insight 1 and 2, we can get

$$\left\|\text{Proj}_{O_M}(\alpha_{iM}V_i)\right\| \propto \left|\Gamma(x_i) \cap \mathcal{C}_\rho^\beta(s)\right|. \quad (15)$$

Eq. 15 shows that the number of intersections between core neurons and the activated neurons in one token reflects how much information this token contains that contributes to the next-token prediction.

From the above analysis, we see that using only the attention score does not consider the angular information. In contrast, core neurons are directly proportional to the projection value, incorporating both the absolute influence and the angular component, thus constituting a more accurate metric. Fig. 5 shows a comparison between using core tokens and using the attention score as metrics. Fig. 8 shows the effectiveness of CoreMatching for different inputs.

Moreover, to the best of our knowledge, this is the first work to investigate the intrinsic relationship between token sparsity and neuron sparsity. **Our findings reveal the inherent correlation between these two sparsity patterns and highlight that the relative angle between tokens is a crucial yet previously overlooked factor in token selection.**

*Table 2.* Comparison with SoTA vision context sparsification methods on vision understanding benchmarks. The best results are bolded. The results of the other methods are from their papers. CoreMatching achieves the best performance in most benchmarks using a smaller number of image tokens. The "Free" indicates whether a method is training-free (i.e., can be applied directly to MLLMs without training.)

| Method | Free | Token | TFLOPs | VQAv2 | GQA | SciQA | TextVQA | POPE | MME | MMB | SEED | VisWiz | MM-Vet |
|---|---|---|---|---|---|---|---|---|---|---|---|---|---|
| LLaVA-1.5-7B | - | 576 | 10.1 | 78.5 | 62.0 | 66.8 | 58.2 | 85.9 | 1510.7 | 64.3 | 66.0 | 50.0 | 31.1 |
| PruMerge+ | ✗ | 146 | 2.5 | 76.8 | 57.4 | 68.3 | 57.1 | 84.0 | 1462.4 | 60.9 | 62.8 | 42.5 | 25.0 |
| FastV | ✓ | 144 | 3.2 | 75.1 | 57.5 | 68.7 | 56.2 | 81.0 | 1458.9 | 63.5 | 62.8 | 47.8 | 26.3 |
| VoCo-LLAMA | ✗ | 128 | 2.2 | 76.9 | 59.8 | - | - | - | - | 61.0 | 59.1 | - | - |
| LLaVA-TRIM | ✗ | 121 | 2.2 | 76.4 | 61.4 | 69.1 | - | 85.3 | 1461.3 | 67.4 | 65.8 | 48.1 | 28.0 |
| Dynamic-LLaVA | ✗ | 115 | 2.5 | 78.0 | 61.4 | 69.1 | **57.0** | 85.0 | 1479.8 | 64.1 | 64.6 | **50.2** | 29.5 |
| **CoreMatching** | ✓ | **64** | **2.1** | **78.5** | 61.9 | 69.5 | 54.5 | **85.9** | 1506.5 | 64.6 | **66.1** | 49.8 | **29.6** |
| LLaVA-1.5-13B | - | 576 | 19.6 | 80.0 | 63.3 | 71.6 | 61.3 | 85.9 | 1531.3 | 67.7 | 68.2 | 53.6 | 36.1 |
| PruMerge+ | ✗ | 146 | 4.9 | 77.8 | 58.2 | 71.0 | 58.6 | 84.4 | 1485.5 | 65.7 | - | 49.7 | 28.0 |
| FastV | ✓ | 144 | 6.0 | 77.0 | 60.1 | 72.8 | 59.0 | 83.2 | 1470.3 | 66.9 | 65.4 | 52.3 | 31.3 |
| LLaVA-TRIM | ✗ | 121 | 5.8 | 75.4 | 59.0 | 72.8 | 54.8 | 86.3 | 1438.0 | 65.7 | 65.9 | 53.2 | 30.3 |
| Dynamic-LLaVA | ✗ | 115 | 4.7 | 78.8 | 62.5 | 72.4 | **59.6** | 86.5 | **1563.3** | 66.9 | 66.5 | 52.8 | 34.8 |
| **CoreMatching** | ✓ | **98** | **4.3** | **79.4** | 63.1 | 72.8 | 58.3 | **87.0** | 1529.9 | **68.5** | 67.4 | 53.2 | **34.9** |

*Table 3.* Hardware performance comparison with the most advanced token sparse and neuron sparse methods. Experiments are performed on a single NVIDIA Titan Xp (12GB). The input is 610 tokens, and the output is 64 tokens. We test the latency of the baseline method based on the LLaVA framework. For PowerInfer, we migrate the predictor provided by Llama2-7b to the LLM module of LLaVA-1.5-7b.

| Method | Cost | Sparsity | | Memory | | | | Pre-filling | | Decoding | |
|---|---|---|---|---|---|---|---|---|---|---|---|
| | Free | Neurons | Token | Kv Cache | FFN | Total | Ratio | TFLOPs | Lat(s) | GFLOPs | Lat(s) |
| LLaVA-7B | ✓ | ✗ | ✗ | 304.8 | 8.26 | 15.63 | 1.00× | 10.1 | 2.23 | 0.845 | 44.2 |
| PruMerge+ | ✗ | ✗ | ✓ | 89.73 | 8.26 | 15.41 | 1.01× | 2.5 | 1.16 | 0.822 | 39.5 |
| FastV | ✓ | ✗ | ✓ | 88.98 | 8.26 | 15.41 | 1.01× | 3.2 | 1.15 | 0.822 | 39.1 |
| Dynamic-LLaVA | ✗ | ✗ | ✓ | 74.40 | 8.26 | 15.39 | 1.01× | 2.5 | 1.15 | 0.820 | 37.6 |
| PowerInfer | ✗ | ✓ | ✗ | 304.8 | 1.83 | 9.20 | 1.70× | 2.2 | 2.03 | 0.516 | 11.6 |
| **CoreMatching** | ✓ | ✓ | ✓ | **48.99** | **1.65** | **8.62** | **1.82×** | **2.1** | **1.05** | **0.486** | **4.81** |

## 4. Experiments

Our experiments are conducted at three levels. First, we evaluate the performance of CoreMatching on various tasks to demonstrate its effectiveness (Sec. 4.1). After that, we deploy CoreMatching on different hardware devices to verify the improvement of hardware performance (Sec. 4.2). Finally, we perform ablation studies to further analyze the effectiveness of each module in CoreMatching (Sec. 4.4). The experimental details are provided in the Appendix D.

### 4.1. Task Performance.

**High Accuracy.** Tab. 2 presents the performance of CoreMatching on various tasks. As shown, CoreMatching demonstrates minimal performance degradation across all tasks and achieves higher accuracy than state-of-the-art baselines on most tasks. Notably, on LlaVA-1.5-7b, Core-Matching achieves lossless performance on datasets such as VQAv2 and GQA while using only about 10% tokens. Furthermore, on SciQA and MMBench, CoreMatching even outperforms the original model. For LlaVA-1.5-13b, Core-Matching demonstrates a clear advantage over other methods, achieving near-lossless accuracy across all datasets.

**Low Cost.** Unlike Dynamic-LLaVA and PruMerge+, which require post-training specific to each model, Core-Matching requires no training or fine-tuning, making it a plug-and-play solution for any model. Additionally, Core-Matching does not require pre-specification of the number of tokens to retain, as it dynamically determines the optimal number of tokens based on the input. Consequently, CoreMatching achieves comparable performance with fewer tokens on average. Specifically, CoreMatching only retains 10% and 17% tokens on average for LlaVA-1.5-7b and LlaVA-1.5-13b, respectively, compared to 20% for PruMerge+ and FastV. Moreover, CoreMatching introduces neuron sparsity in addition to token sparsity, further reducing computational requirements during the decoding stage.

### 4.2. Hardware Performance

**Reduced Inference Memory.** Tab. 3 compares the hardware resources required for inference between CoreMatching and other sparse inference methods on an NVIDIA Titan Xp (12GB). CoreMatching can simultaneous sparsity across both token and neuron dimensions, CoreMatching significantly reduces the memory footprint of the KV cache during inference as well as the memory required for neurons during

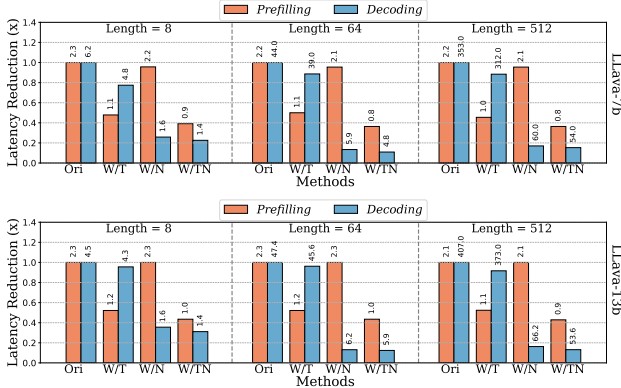

*Figure 9.* Latency comparison of token-only/neurons-only/both sparse on NVIDIA TiTAN Xp. W/T means only core tokens, W/N means only core neurons, and W/TN means CoreMatching. The number on the bar means how many seconds it took.

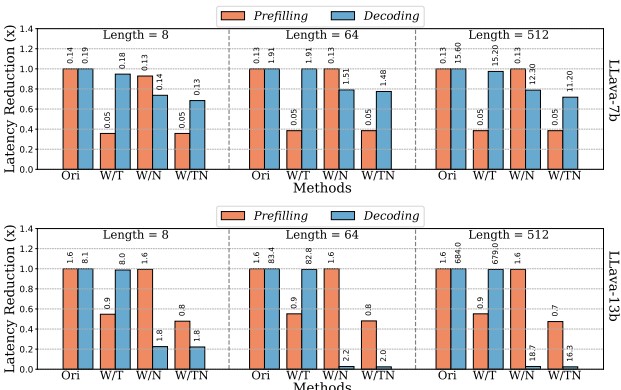

*Figure 10.* Latency comparison of token-only/neurons-only/both sparse on NVIDIA RTX 6000.

decoding. When the input token length is short, CoreMatching requires only about half the memory of the original model during decoding, eliminating memory-bound limitations on resource-constrained devices and significantly reducing inference latency. For long input token sequences, CoreMatching also can effectively reduce primary memory usage by minimizing the KV cache size.

**Reduced Inference Latency.** In contrast to methods that achieve partial acceleration, such as token sparsity for pre-filling time reduction or neuron sparsity for decoding time reduction, CoreMatching delivers comprehensive inference acceleration through multidimensional sparsity. As illustrated in Tab. 3, CoreMatching achieves a 2.1× speedup in the pre-filling stage and a 9.2× speedup in the decoding.

### 4.3. Generalization Analysis

**Task Generalization.** To verify the task generalization of our method, we conducted extensive evaluations on LLaVA-1.5-7B and LLaVA-1.5-13B across additional OCR and chart/document understanding tasks. We also reproduced

*Table 4.* Peformance comparison with popular acceleration methods on OCR, chart, and document understanding tasks. The basic models are LLaVA-1.5-7B and LLaVA-1.5-13B. All tasks are evaluated on test set. "TF" indicates TFLOPs. "OCRB" indicates OCRBench

| Method | Token | TF | DocVQA | InfoVQA | ChartQA | OCRB | AI2D |
|---|---|---|---|---|---|---|---|
| LLaVA-1.5-7B | 576 | 10.1 | 22.2 | 22.3 | 18.2 | 31.3 | 55.2 |
| PreMerge+ | 146 | 2.5 | 17.9 | 21.1 | 15.8 | 26.2 | 47.3 |
| FastV | 144 | 3.2 | 20.6 | **22.7** | 16.2 | 28.1 | 51.2 |
| **CoreMathing** | 48.4 | **2.1** | **21.9** | **22.7** | 17.8 | **30.5** | **55.2** |
| LLaVA-1.5-13B | 576 | 19.6 | 24.6 | 25.1 | 18.2 | 33.6 | 59.2 |
| PreMerge+ | 146 | 4.9 | 14.6 | 23.6 | 16.9 | 27.8 | 49.1 |
| FastV | 144 | 6.0 | 22.9 | 24.5 | 17.1 | 30.1 | 54.9 |
| **CoreMating** | **56.3** | **4.3** | **23.4** | **24.8** | 17.5 | **33.5** | **59.3** |

*Table 5.* Performance of CoreMatching on the Qwen2.5-VL-3B and Qwen2.5-VL-7B models on OCR, chart, and document understanding tasks. The results of the original model are extracted from the Qwen2.5-VL technical article. Since the number of image tokens in Qwen2.5-VL is dynamic, we take the average of different tasks as the number of tokens used. "Neur" indicates Neurons.

| Model | Tokens | Neur | DocVQA | InfoVQA | ChartQA | OCRB | AI2D |
|---|---|---|---|---|---|---|---|
| Claude-3.5 Sonnet | - | - | 95.2 | 74.3 | 90.8 | 78.8 | 81.2 |
| Gemini 1.5 Pro | - | - | 93.1 | 81.0 | 87.2 | 75.4 | 88.4 |
| GPT 4o | - | - | 91.1 | 80.7 | 86.7 | 73.6 | 84.6 |
| Qwen2.5-VL 3B | 235.2 | 11008 | 93.9 | 77.1 | 84.0 | 79.7 | 81.6 |
| **CoreMatching** | 24.1 | 4403 | 93.2 | 77.1 | 83.6 | 79.2 | 81.6 |
| Qwen2.5-VL 7B | 234.9 | 11008 | 95.7 | 82.6 | 87.3 | 86.4 | 83.9 |
| **CoreMatching** | 23.8 | 4403 | 94.8 | 82.3 | 87.0 | 85.6 | 84.2 |

and collected results for FastV and PruMerge on these tasks to ensure a fair comparison. As shown in Tab. 4, CoreMatching consistently outperforms other acceleration methods on most tasks while incurring only minimal performance degradation compared to the original models. Notably, for tasks such as DocVQA and InfoVQA, which often require understanding only small portions of text within an image, CoreMatching's dynamic token pruning allows it to achieve near-lossless performance using significantly fewer tokens—on average, less than 10% of the original token.

**Model Generalization.** To further evaluate the generalizability and effectiveness of our method on more advanced architectures, we conducted comprehensive experiments on Qwen2.5-VL-3B and Qwen2.5-VL-7B, two representative LVLMs that incorporate dynamic resolution mechanisms to adaptively allocate computational resources based on visual content complexity. These models present a more challenging evaluation setting due to their sophisticated token selection and multi-scale processing capabilities. As summarized in Tab. 5, CoreMatching continues to exhibit robust performance under this setting. Notably, it achieves near-lossless performance while utilizing only approximately 10% of the visual tokens, demonstrating its capability to distill and preserve essential visual information with high efficiency. Even more, on the AI2D dataset, CoreMatching not only retains accuracy but actually surpasses the performance of the original full-token model, underscoring its adaptability and potential to enhance model performance through better

*Table 6.* Performance comparison of different LVMs on video reasoning benchmarks. The base model for all acceleration methods is Video-LLaVA. Results of PruMerge and FastV are taken from the original papers.

| Methods | Size | MSVD-QA | | MSRVT-QA | | ActivityNet-QA | |
|---|---|---|---|---|---|---|---|
| | | Accuracy | Score | Accuracy | Score | Accuracy | Score |
| FrozenBiLM | 1B | 32.2 | - | 16.8 | - | 24.7 | - |
| VideoChat | 7B | 56.3 | 2.8 | 45.0 | 2.5 | - | 2.2 |
| LLaMA-Adapter | 7B | 54.9 | 3.1 | 43.8 | 2.7 | 34.2 | 2.7 |
| Video-LLaMA | 7B | 51.6 | 2.5 | 29.6 | 1.8 | 12.4 | 1.1 |
| Video-ChatGPT | 7B | 64.9 | 3.3 | 49.3 | 2.8 | 35.2 | 2.7 |
| Video-LLaVA | 7B | 70.7 | 3.9 | 59.2 | 3.5 | 45.3 | 3.3 |
| FastV | 7B | 71.0 | 3.9 | 57.0 | 3.5 | - | - |
| PruMerge | 7B | 71.1 | 3.9 | 58.4 | 3.5 | 48.3 | 3.4 |
| PruMerge+ | 7B | 71.1 | 3.9 | 59.3 | **3.6** | 47.7 | **4.4** |
| **CoreMatching** | **7B** | **71.6** | **3.9** | **59.8** | **3.6** | **48.4** | 3.4 |

*Table 7.* The average number of core tokens for different tasks.

| Type | Multi Choice | Questions and Answers | | | | | | Long Text Generate |
|---|---|---|---|---|---|---|---|---|
| Tasks | SciQA | VQAv2 | GQA | VisWiz | TextVQA | POPE | MME | MM-Vet |
| Number | 27.9 | 63.5 | 76.6 | 76.9 | 64.1 | 71.1 | 66.9 | 93.9 |

token selection. These results collectively validate that CoreMatching remains effective when applied to state-of-the-art LVLMs with dynamic resolution, reinforcing its potential as a general-purpose visual token reduction strategy that scales across model sizes and architectural paradigms.

**Performance on Video Task.** To verify the scalability of CoreMatching on Video tasks, we used Video-LLaVA as the base model and conducted experiments on multiple video tasks including MSVD-QA, MSRVT-QA, and ActivityNet-QA. As shown in Tab. 6, CoreMatching consistently outperforms existing advanced acceleration methods across multiple video reasoning benchmarks. Specifically, on MSRVT-QA, CoreMatching reaches an accuracy of 59.8%, outperforming origianl Video-LLaVA (59.2%). On ActivityNet-QA, CoreMatching improves accuracy (48.4%) compared to the origianl Video-LLaVA (45.3%). This suggests that token redundancy in video frames can be substantial, and strategic pruning not only reduces computational overhead but may also enhance model performance by eliminating noisy or irrelevant tokens. These results collectively highlight the strong potential of CoreMatching as a general-purpose acceleration framework for video LLMs.

### 4.4. Ablation Study

**Module Ablation.** To further analyze the contributions of token sparsity and neuron sparsity to inference acceleration, we conducted ablation experiments on various hardware. As shown in Fig. 9 and 10, we compared the acceleration achieved by token sparsity alone, neuron sparsity alone, and CoreMatching (sparsity applied to both tokens and neurons) during the pre-filling and decoding stages. Across different devices, models, and output token lengths, CoreMatching consistently achieved significant acceleration, with more

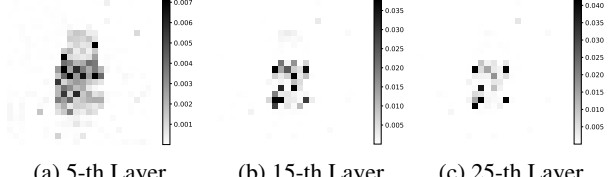

| (a) 5-th Layer | (b) 15-th Layer | (c) 25-th Layer |
|---|---|---|

*Figure 11.* Number of tokens required at different layers.

pronounced gains as the output token length increased.

Token-only sparsity primarily accelerated the pre-filling stage, with minimal impact on the decoding stage (limited to reducing KV computation). Conversely, neuron-only sparsity mainly accelerated the decoding stage, offering little improvement during the pre-filling stage. By simultaneously reducing memory and computational load, CoreMatching achieved higher acceleration rates in both stages.

**Core Tokens Across Different Tasks.** Due to the input-adaptive of CoreMatching, it can automatically determine how many tokens to retain based on the data distribution of each task. Tab. 7 shows the number of core tokens retained for different task types. As observed, CoreMatching adaptively adjusts the required tokens according to task difficulty. For instance, simpler tasks such as multiple-choice question answering (e.g., SciQA where one must choose between options A or B) require only 27.9 tokens on average. In contrast, more challenging tasks involving longer text generation (e.g., MM-Vet, which requires generating medical records from images) need around 93.9 tokens on average. Notably, even for the most difficult tasks, CoreMatching needs fewer than 20% of tokens, outperforming other token sparsity methods in terms of token reduction.

**Tokens Required at Different Layers.** Another crucial aspect of token sparsity is understanding how many tokens are needed at each layer. Using our proposed optimal evaluation metric, we visualize and analyze the token requirements across different layers. As illustrated in Fig. 11, image tokens that are closely correlated with text tokens consistently exhibit higher projection values, and as the layer depth increases, the number of tokens with high projection values gradually decreases. This suggests that the model focuses on a larger number of tokens in the earlier layers, and more tokens can be pruned in later layers, aligning with results from previous studies.

## 5. Conclusion

This paper introduces CoreMatching, a co-adaptive sparse inference framework that accelerates VLMs by leveraging the matching relationship between core neurons and core tokens. Additionally, we also propose a more principled token measurement criterion and theoretically derive how activation layers influence token interactions, providing new insights into how VLMs process image and text information.

## Acknowledgements

Qinsi Wang, Jianyi Zhang and Yiran Chen disclose the support from NSF 2112562, ARO W911NF-23-2-0224, and NAIRR Pilot project NAIRR240270. We thank area chair and reviewers for their valuable comments.

## Impact Statement

This paper aims to advance the field of adaptive sparse inference for VLMs. We believe our work has significant potential for applications in deploying large models on resource-constrained devices. While our research may have various societal implications, we do not find any particular aspect that requires special emphasis in this context.

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

**Organization** In this appendix, we provide in-depth descriptions of the materials that are not covered in the main paper, and report additional experimental results. The document is organized as follows:

## A. Related Work

### A.1. Activation Sparsity

In a single Feed-Forward Network (FFN) block in LLMs, there are typically two linear layers, $W_u, W_d \in \mathbb{R}^{N \times 4N}$. For a single token, denote the input representation of the FFN layer as $x$, the output $y$ can be expressed as:

$$A = \sigma(xW_u), \quad y = AW_d \tag{16}$$

where $\sigma$ represents the activation function, such as ReLU or SiLU. Intermediate output $A = [a_1, a_2, \ldots, a_{4N}]$, where $a_n$ is the activation value of the $n$-th neuron.

Previous work (Song et al., 2023; Xue et al., 2024; Wang & Zhang, 2023; Wang et al., 2023; Lin et al., 2025) indicates that individual tokens in LLMs exhibit significant activation sparsity. For example, in the OPT-30B, a single token activates only approximately 10% of the neurons (Alizadeh et al., 2023). Therefore, if the activated neurons can be accurately predicted in advance, a substantial amount of activation computation can be eliminated, thereby accelerating model inference without compromising performance. This potential has attracted considerable attention from researchers. For example, DejaVu (Liu et al., 2023) inserts an MLP predictor into each FFN block to forecast which neurons will be activated, achieving a prediction accuracy of 93%. PowerInfer (Song et al., 2023) introduces the concepts of hot neurons and cold neurons to respectively represent frequently and rarely activated neurons, and accelerates inference by intelligently allocating hardware resources. LLM in Flash (Alizadeh et al., 2023) and PowerInfer2 (Xue et al., 2024) optimize this algorithm for mobile devices, thereby reducing the DRAM requirements for LLM inference on mobile phones.

Notably, recent work CoreInfer (Wang et al., 2024) proposed a sentence-level adaptive activation sparsity inference method without the need of predictors. CoreInfer identifies a set of core neurons that most frequently and strongly activated for each input sentence. Experiments demonstrated that for a given input sentence, LLMs require only a static set of core neurons to maintain performance. Leveraging sentence-level sparsity, CoreInfer eliminates the need for frequent neuron switching during the decoding stage, achieving a $10.33\times$ speedup on NVIDIA Titan Xp GPU with negligible performance loss.

The effectiveness of CoreInfer highlights the ability of core neurons to capture the most critical information of the input, underscoring their remarkable potential. In this paper, we further explore the characteristics of core neurons in VLMs, and propose a novel one-pass co-adaptive sparsity inference.

### A.2. Token Sparsity

The notorious quadratic complexity in Transformers is a well-known issue and one of the key bottlenecks in scaling input sequence lengths. In VLMs, this problem becomes even more pronounced as the length of input image tokens increases. To address this challenge, a multitude of adaptive token sparsity methods have been proposed (Huang et al., 2024)(Ye et al., 2024)(Shang et al., 2024)(Chen et al., 2025). For instance, Prumerge (Shang et al., 2024) observed the sparsity in the distribution of attention scores between class token and visual tokens, and employed attention scores as an evaluation metric to determine which tokens should be discarded. FastV (Chen et al., 2025) further explored and demonstrated the inefficiency of visual attention in VLMs, proposing a plug-and-play approach that achieved a 45% reduction in FLOPs on Llava-1.5-13b. These methods validate that VLMs require only a small subset of important tokens to achieve nearly lossless performance.

However, the acceleration and efficiency gains from purely token-level sparsity are limited, especially since the speedup benefits during the decoding phase are significantly diminished. In this paper, we thoroughly investigate the intrinsic relationship between token sparsity and neuron sparsity and propose a co-adaptive sparsity method. Our method achieves comprehensive acceleration at nearly zero cost.

# B. Algorithm

The algorithm of CoreMatching is shown in Algorithm 1. In the pre-filling stage, we calculate core neurons in each FFN block and calculate core tokens in the $l$-th layer. After calculating the core tokens, we only use the core tokens in the subsequent inference process. In the decoding stage, we only use the retained core neurons for calculation.

---

**Algorithm 1** CoreVison: Co-adaptive sparse inference

---

**Input:** Sentence $s = [x_1, x_2, ..., x_M]$; Token-pruning layer $\mathcal{L}$,
        Sparsity hyperparameters $\rho$, $\beta$.

**Step1: Pre-filling Stage**

**for** *layer* $l = 1, 2, ..., L$ **do**
     Capture the activation of all tokens;

     **for** *token* $x = x_1, x_2, ..., x_M$ **do**
         Record neurons with the largest $\rho\%$ activation as token-wise core neurons, $\mathcal{C}_\rho(x)$.
     **end**

     Record the most frequent $\beta\%$ neurons in $\{\mathcal{C}_\rho(x_1), \mathcal{C}_\rho(x_2), ..., \mathcal{C}_\rho(x_M)\}$ as sentence-wise core neurons $\mathcal{C}_\rho^\beta(s)$;

     Keep only core neurons $\mathcal{C}_\rho^\beta(s)$ on the device.

     **if** $l = \mathcal{L}$ **then**
         Record the neurons activated by each token $\Gamma(x)$;
         Calculate the number of intersections $\left|\Gamma(x) \cap \mathcal{C}_\rho^\beta(s)\right|$;
         Use maximum geometric distance to get threshold $T_k$;
         Select all $x_i$ that $\left|\Gamma(x_i) \cap \mathcal{C}_\rho^\beta(s)\right| \geq T_k$ as core tokens;
         Keep only core tokens for inference.
     **end**
**end**

**Step2: Decoding Stage**

**for** *layer* $l = 1, 2, ..., L$ **do**
     Use $\mathcal{C}_\rho^\beta(s)$ kept in the pre-filling stage for inference.
**end**

---

# C. Assumption Explanation

In this section, we provide more comprehensive experimental proofs and discussions for the two observations proposed.

For **Observation 1**, we show $W_Q @ W_K.T$, $W_V @ W_V.T$, $W_D @ W_D.T$ of all different layers of LLaVA-1.5-7B in Fig. 12, 13, and 14. It can be seen that different layers have this orthogonal relationship.

In fact, the orthogonal relationship of matrices in neural networks has been studied since a long time ago. In particular, (Li et al., 2019) proposed a new regularization method that encourages the weight matrix of the neural network to maintain orthogonality during training by introducing a self-orthogonality module. This method helps to improve the training stability and generalization ability of the model. (Bansal et al., 2018) explores adding orthogonal regularization to weights during training to improve training stability. The author proposed an orthogonal regularization method for weights, aiming to solve the gradient vanishing and explosion problems encountered by deep convolutional neural networks during training.

It can be seen that modules with orthogonality are found in various different models to improve the training stability and performance of the model. To the best of our knowledge, we are the first work to intuitively show this orthogonal performance in LLM, which can be more fully explored in subsequent research.

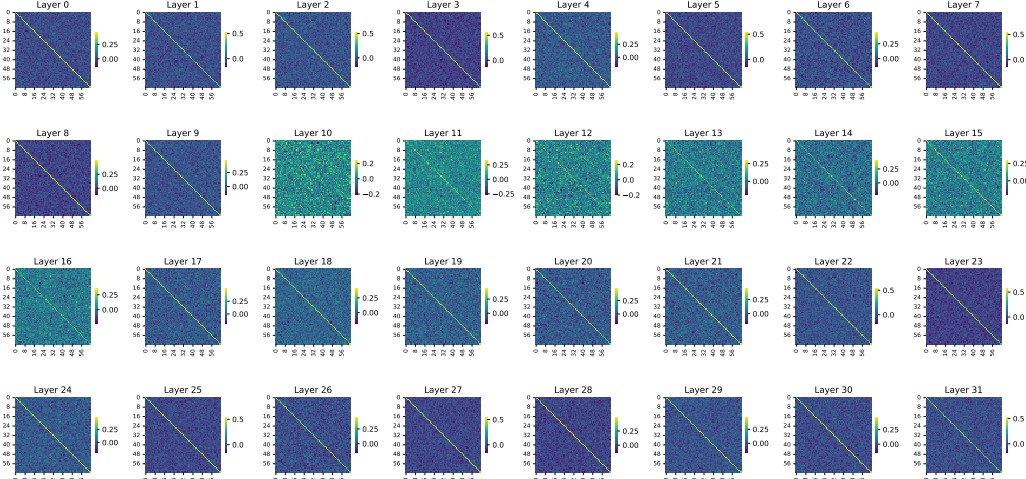

*Figure 12.* Visualization of $W_Q @ W_K.T$ at different layers in LLaVA-1.5-7b.

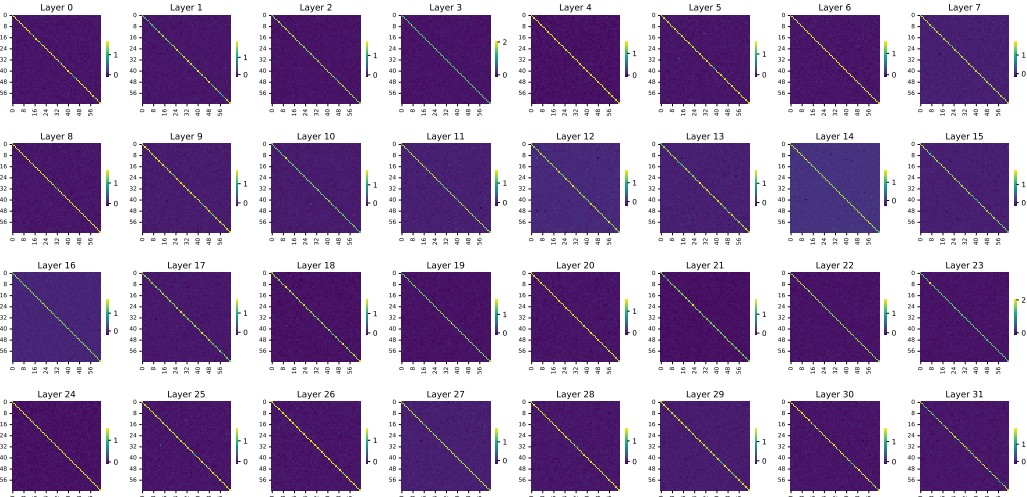

*Figure 13.* Visualization of $W_D @ W_D.T$ at different layers in LLaVA-1.5-7b.

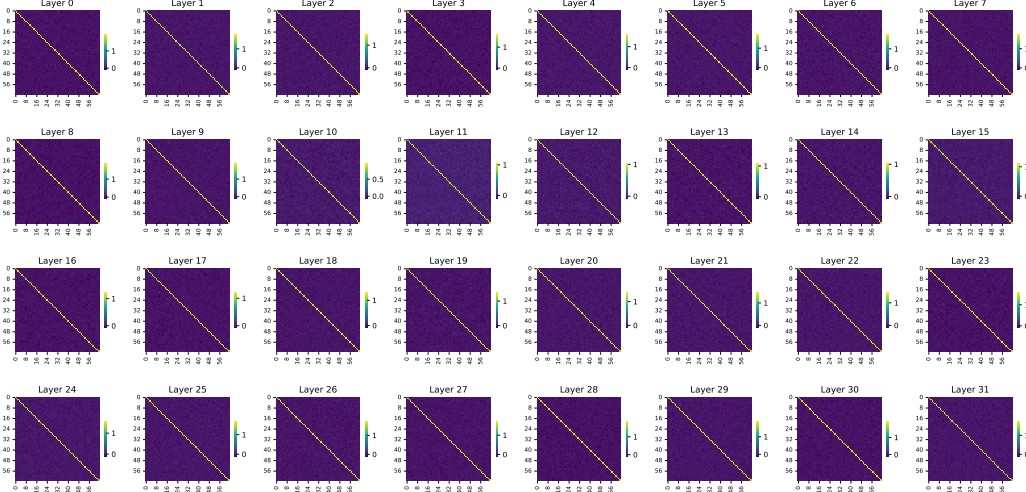

*Figure 14.* Visualization of $W_V @ W_V.T$ at different layers in LLaVA-1.5-7b.

For **Observation 2**, we show the distribution of 32 layers of LLaVA-1.5-7b in Fig. 15. It can be seen that starting from the 4-th layer, $cos(\angle(A_i, A_M))$ is proportional to the number of co-activated neurons. This is actually in line with intuition. On the one hand, the more neurons $A_i$ and $A_M$ co-activate, the more non-zero values they have in the same positions, and the larger the product. On the other hand, $A_i$ and $A_M$ have positive values in more of the same dimensions, indicating that their directions are closer.

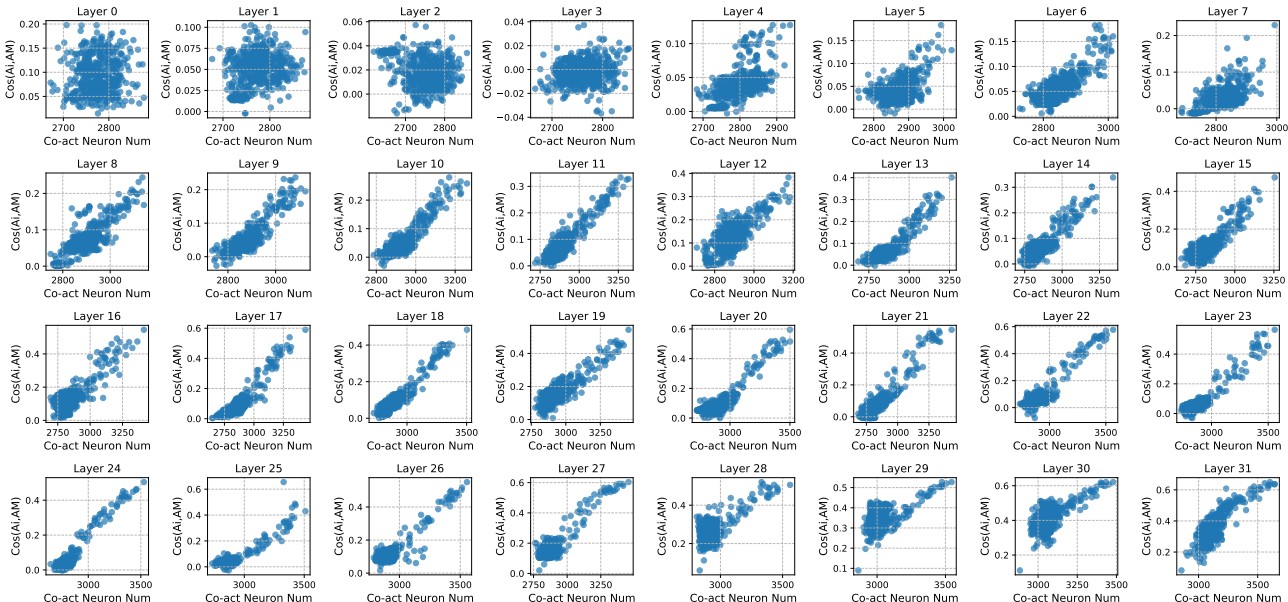

*Figure 15.* Visualization of $cos(\angle(A_i, A_M))$ and co-act neurons number at different layers in LLaVA-1.5-7b.

For proof of $\cos(\angle(y_i, y_M)) = \cos(\angle(A_i, A_M))$, $\cos(\angle(y_i, y_M))$ can be written as

$$\begin{aligned}
\cos(\angle(y_i, y_M)) &= \langle A_i W_{\rm d}, A_M W_{\rm d}\rangle/(\|y_i\|\|y_M\|) \\
&= A_i(W_{\rm d}W_{\rm d}^T)A_M^T/(\|y_i\|\|y_M\|) \\
&= \eta\langle A_i, A_M\rangle /(\|y_i\|\|y_M\|),
\end{aligned} \tag{17}$$

where $\eta$ is a constant based on Observation 1. Furthermore, since $W_d$ is an orthogonal matrix, we have

$$\begin{aligned}
\|y_i\|^2 = \|A_i W_d\|^2 &= (A_i W_d)(A_i W_d)^T \\
&= A_i(W_d W_d^T)A_i^T = \eta A_i A_i^T = \eta\|A_i\|^2.
\end{aligned} \tag{18}$$

which means $\|y_i\| = \sqrt{\eta}\|A_i\|$. Substituting this into Eq. 17 we can have

$$\begin{aligned}
\cos(\angle(y_i, y_M)) &= \eta\langle A_i, A_M\rangle /(\eta\|A_i\|\|A_M\|) \\
&= \cos(\angle(A_i, A_M))
\end{aligned} \tag{19}$$

This shows that the orthogonal matrix $W_d$ does not change the angles between the input vectors.

## D. Experiments Settings

**Models and Tasks.** Our main experiments are conducted on LlaVA-1.5-7b and LlaVA-1.5-13b, using FP16 for all models. Following the same evaluation setting as LlaVA, we evaluate our methods on ten classical tasks, including VQAv2 (Goyal et al., 2017), GQA (Hudson & Manning, 2019), VizWiz (Gurari et al., 2018), SciQA (Lu et al., 2022), TextVQA (Singh et al., 2019), POPE (Li et al., 2023b), MMBench (en) (Liu et al., 2025a), SEED (image) (Li et al., 2023a), and MM-Vet (Yu et al., 2023). We also provide additional results on more models in the Appendix.

**Hardware.** We conduct experiments on two distinct hardware configurations. NNVIDIA Quadro RTX 6000 (24GB), representing high-performance hardware scenarios. In contrast, NVIDIA TITAN XP GPU (12G), representing low-performance hardware scenarios. We also provide experimental results on more hardware devices in the Appendix.

**Baselines.** Given the multi-dimensional sparsity of CoreMatching, we compare our method against approaches using either token-level sparsity or neuron-level sparsity. For token sparsity, we compare with state-of-the-art token pruning methods such as PruMerge+, FastV, VoCo-LLaMA, LLaVA-HiRED, LLaVA-TRIM, and Dynamic-LLaVA. For neuron-level sparsity, since there has been no prior work specifically on activation sparsity in VLMs, we adapt the predictor from PowerInfer to Llava to emulate the time required for MLP-based activation sparsity on Llava.

**Implementation Details.** CoreMatching uses the same hyperparameter settings across all activation layers for all models. For the computation of core neurons, we follow the hyperparameter setting from CoreInfer with $\rho = 0.2, \beta = 0.4$. For core tokens, we set $l = 2$, consistent with FastV.

# E. Additional Experiments

We supplement here the experimental results that are not included in the main text.

## E.1. Latency on NVIDIA A100

Fig. 16 shows the latency comparison on NVIDIA A100. It can be seen that CoreMatching can achieve excellent acceleration effects even without memory limit. And as the batch size increases, the acceleration effect becomes more and more obvious.

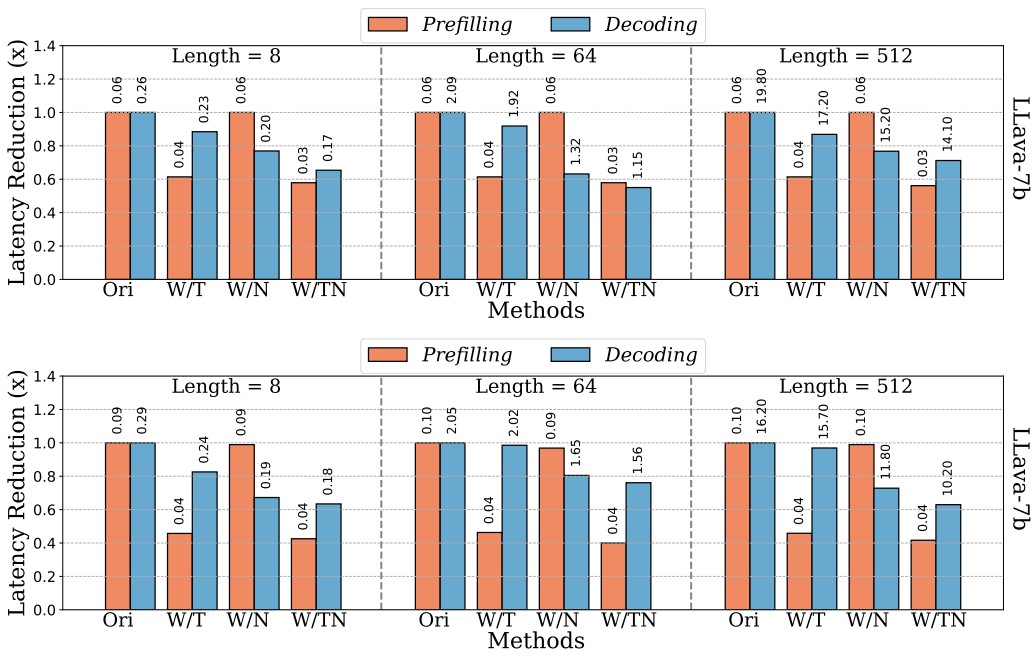

*Figure 16.* Latency comparison of token-only/neurons-only/both sparse on NVIDIA A100.

## E.2. Experimental Results on LLaMA3.2

We show the experimental results of CoreMatching on LLaVA in the main text. In order to demonstrate the model generalization of CoreMatching, we further show the experimental results on LLaMA3.2 here. Since LLaMA-3.2 uses the crossAttention layer, the activation layer cannot be used to guide the token pruning of attention. We directly perform activation sparseness without token sparseness. The experimental results are shown in Tab. 8. For LLaMA3.2-11B, CoreMatching can still achieve near-lossless performance. This shows the applicability of CoreMatching to models of different architectures.

*Table 8.* Experimental results of CoreMatching on LLaMA-3.2-11b.

| Methods | TextVQA | MME | VQAV2 | VisWiz | POPE | GQA |
|---------|---------|--------|-------|--------|------|------|
| Number | 69.7 | 1384.5 | 81.4 | 52.5 | 88.7 | 68.6 |
| Number | 66.5 | 1381.2 | 79.8 | 49.8 | 87.8 | 66.2 |

# F. Visualization of Results

In this section, we visualize the experimental results of CoreMatching. As shown in Fig. 17, for different inputs, CoreMatching can always retain the tokens that are important to the output.

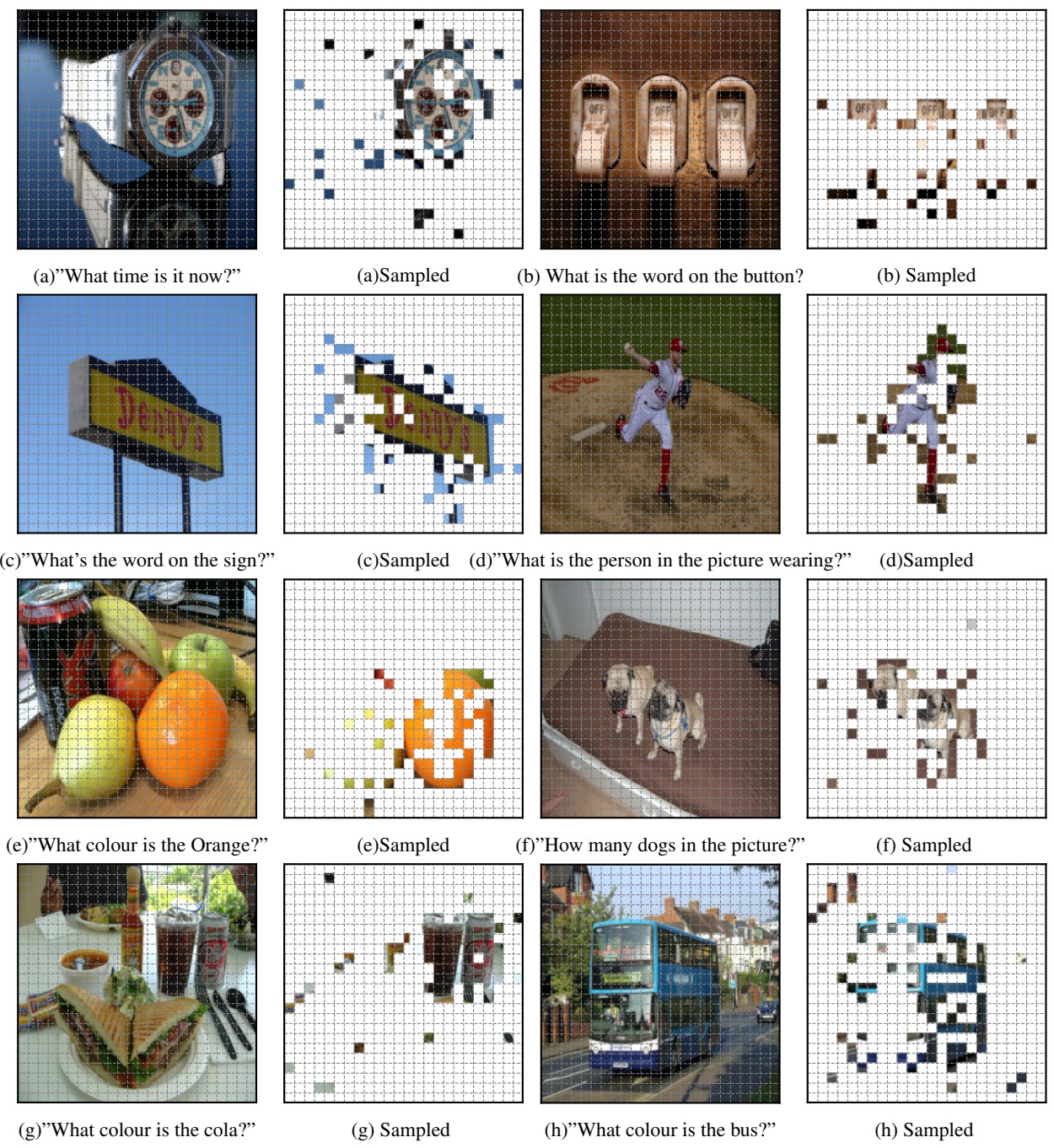

(a)"What time is it now?"     (a)Sampled     (b) What is the word on the button?     (b) Sampled

(c)"What's the word on the sign?"     (c)Sampled     (d)"What is the person in the picture wearing?"     (d)Sampled

(e)"What colour is the Orange?"     (e)Sampled     (f)"How many dogs in the picture?"     (f) Sampled

(g)"What colour is the cola?"     (g) Sampled     (h)"What colour is the bus?"     (h) Sampled

*Figure 17.* Examples of CoreMatching sampled tokens for different inputs.

