# OpenReview forum: "CoreMatching: A Co-adaptive Sparse Inference Framework with Token and Neuron Pruning for Comprehensive Acceleration of Vision-Language Models"
_ICML.cc/2025/Conference — ICML 2025 poster_

### Official Review · Reviewer_cD4w · 2025-03-12

**Overall Recommendation:** 3

**Summary:**

CoreMatching combines token sparsity and neuron sparsity through the interaction between core neurons and core tokens, achieving comprehensive acceleration of VLMs. The proposed projection-guided criterion provides a more accurate way to evaluate token importance, and the co-adaptive framework effectively reduces computational costs while maintaining high performance, which has important practical significance for deploying VLMs on resource-constrained devices.

**Claims And Evidence:**

The claims in this paper should be correct.

**Essential References Not Discussed:**

Token pruning and merging are two critical technical paths for token sparsity. While the authors have thoroughly compared pruning-based methods, there may be a lack of comparison against merging-based methods, _e.g._, [1,2].

[1] Daniel Bolya, Cheng-Yang Fu, Xiaoliang Dai, Peizhao Zhang, Christoph Feichtenhofer, and Judy Hoffman. Token merging: Your vit but faster. In International Conference on Learning Representations, 2023.

[2] Chen Ju, Haicheng Wang, Haozhe Cheng, Xu Chen, Zhonghua Zhai, Weilin Huang, Jinsong Lan, Shuai Xiao, and Bo Zheng. Turbo: Informativity-driven acceleration plug-in for vision-language large models. In European Conference on Computer Vision, pages 436–455. Springer, 2025.

**Experimental Designs Or Analyses:**

My main concerns are about experimental verification.
1. **Compatibility with flash-attention**: Is this technique compatible with flash-attention? As an acceleration-focused work, it should align with mainstream acceleration frameworks. I did not see any discussion from the authors about this.

2. **Necessity of Core Neurons**: The paper claims "The angle should be an essential indicator for token importance evaluation". While I agree with this claim, why not directly integrate the angle and attention score instead of using core neurons to guide core token selection? Is there experimental evidence supporting the superiority of this design choice?

3. **Insufficient Validation of Generalization**:
(1) **Video datasets**: Many existing methods report performance on video datasets, _e.g._, Video MME, but CoreMatching does not provide such results.
(2) **LVLMs using dynamic resolution**: LLaVA-1.5 does not use dynamic resolution. How does CoreMatching perform on LVLMs using dynamic resolution (e.g., Qwen2.5-VL or LLaVA-NEXT)?

4. **Applicability to Training Acceleration**: Can CoreMatching be extended to accelerate training?

**Methods And Evaluation Criteria:**

The used evaluation criteria is standard in token pruning / merging. The experimental results are impressive.

**Other Comments Or Suggestions:**

n/a

**Other Strengths And Weaknesses:**

n/a

**Questions For Authors:**

Please refer to comments about experiments and references. In addition, will you open-source the code after acceptance?

**Relation To Broader Scientific Literature:**

n/a

**Theoretical Claims:**

The theoretical claims seem to be solid.

---

> ### Author Rebuttal · Authors · 2025-04-01
>
> Dear Reviewer cD4w,
>
> We would like to thank you for taking the time to review our paper and provide valuable feedback. We appreciate the opportunity to address your questions and concerns. We believe these discussions and revisions will help further improve the paper.
>
> ***Q1. Is this technique compatible with FlashAttention?***
>
> **A1.** Thank you for your valuable question. **Our method is fully compatible with FlashAttention**, as CoreMatching sparsifies in the FFN block, orthogonally complementing FlashAttention in the attention block. *Our implementation already leverages FlashAttention* by default through the LLaVA-1.5 framework, and we will clarify this explicitly in the revision.
>
> ***Q2. Why not directly use angle instead of core neurons to guide core token selection?***
>
> **A2.** Thank you for your insightful comments.
>
> 1. **Theoretically, angle-based metrics alone may cause conflicts when combined directly with neuron sparsity, due to inconsistent token-neuron interactions.** Our neuron-guided token sparsification ensures consistency—core tokens activate core neurons, stabilizing neuron activation and preventing negative interference.
> 2. **Empirically, our new experiments (Tab. 4, provided link: https://drive.google.com/file/d/1tYVMhd12V6Qgx6aqyLEK1rG8lCPOJvcb/view?usp=sharing) demonstrate improved task performance from combining neuron-guided token selection with neuron sparsity.** Using angular metrics without alignment indeed degraded performance, supporting our theoretical insights. We agree angle-based metrics remain valuable for token sparsity and will expand this discussion accordingly.
>
> We will expand this discussion in the revised paper and thanks for your question.
>
> ***Q3. More experiments on video datasets.***
>
> **A3.** Thank you for the constructive suggestion. We have conducted additional experiments on video datasets, and the results are presented in Tab. 3 of the anonymous link below:
> https://drive.google.com/file/d/1tYVMhd12V6Qgx6aqyLEK1rG8lCPOJvcb/view?usp=sharing
>
> As shown, CoreMatching continues to outperform advanced acceleration methods such as FastV and PruMerge on video tasks. Interestingly, due to the high redundancy in video frames, token pruning sometimes even improves performance over the original model. This highlights CoreMatching’s strong potential in accelerating video LLMs. We will include these results in the revised version. Thank you again for your valuable suggestion.
>
> ***Q4. How does CoreMatching perform on LVLMs using dynamic resolution?***
>
> **A4.** Thank you for the helpful question. We have added experiments on Qwen2.5-VL across OCR, chart, and document understanding benchmarks. These results, along with additional experiments on LLaVA, are shown in **Tab.2** of the anonymous link:
> https://drive.google.com/file/d/1tYVMhd12V6Qgx6aqyLEK1rG8lCPOJvcb/view?usp=sharing
>
> As shown, CoreMatching achieves near-lossless performance even after pruning approximately 90% of the tokens, demonstrating its effectiveness in state-of-the-art LVLMs with dynamic resolution.  We will include these results in the revised version. Thank you again for your thoughtful suggestion.
>
> ***Q5. Can CoreMatching be extended to accelerate training?***
>
> **A5.** Thank you for the constructive question. We believe CoreMatching can significantly accelerate training, with substantial potential:
>
> - **Forward pass:** CoreMatching enhances efficiency in both pre-filling and decoding, benefiting supervised fine-tuning and RL-based training (e.g., addressing decoding bottlenecks in RL methods like GRPO).
>
> - **Backward pass:** CoreMatching reduces computational costs during gradient updates by activating only core neurons (thus updating fewer weights) and pruning tokens (reducing memory usage and computation).
>
> We plan to explore this in future work and will add relevant discussion in the revised version.
>
> ***Q6. Essential references not discussed: lack of comparison against merging-based methods.***
>
> **A6.** Thank you for pointing out this important omission. We are happy to provide the following discussion:
>
> In fact, CoreMatching is complementary to merging-based approaches. On one hand, the angle similarity and neuron activation metrics proposed in CoreMatching can serve as alternative importance or similarity scores in merging-based methods, replacing conventional attention-based metrics. On the other hand, token merging strategies can also be incorporated into CoreMatching to better retain globally informative tokens for complex tasks.
>
> We will cite the suggested works and add a discussion in the revised version. Thank you again for the valuable recommendation.
>
> ***Q7. Will you open-source the code after acceptance?***
>
> **A7.** Thank you very much for your interest in our work. **Yes —** **we fully intend to release the code upon acceptance**.
>
> Sincerely,
>
> Authors

---

### Official Review · Reviewer_T5zD · 2025-03-14

**Overall Recommendation:** 3

**Summary:**

The authors explore a fundamental question on jointly leveraging token and neural sparsity to enhance the inference efficiency of vision-language models (VLMs). The paper introduces the concept of core neurons and investigates their correspondence with core tokens. Building on this observation, the authors propose CoreMatching, a co-adaptive sparse inference framework that exploits the interplay between token and neuron sparsity to accelerate VLMs. Experimental results demonstrate that the proposed method outperforms state-of-the-art baselines across ten image understanding tasks and three hardware platforms.

## update after rebuttal
I appreciate the authors’ detailed responses, which have successfully addressed my concerns.

**Claims And Evidence:**

Yes

**Essential References Not Discussed:**

No

**Experimental Designs Or Analyses:**

Yes

**Methods And Evaluation Criteria:**

Yes

**Other Comments Or Suggestions:**

Please see the questions.

**Other Strengths And Weaknesses:**

The paper is well-motivated, addressing an interesting, fundamental, and important research question on VLM efficiency. It is well-structured, beginning with preliminary concepts and systematically deriving the relationship between core neurons and core tokens. The experimental results are compelling, demonstrating the effectiveness of the proposed approach.

**Questions For Authors:**

1. In the proposed method, the authors retain the top $\rho\%$ most activated neurons as token-wise core neurons and identify the most frequent $\beta\%$ neurons as sentence-wise core neurons. A key question is how to determine these two hyperparameters to effectively balance token sparsity and neuron sparsity.

2. The proposed method focuses solely on core neurons in the FFN block. Could attention mechanisms also be incorporated to further enhance efficiency?

3. In the right column of L128, “3%” should be “4.6%”.

4. I am curious about the choice of hardware for the throughput experiments. Why did the authors use the NVIDIA Titan XP (12GB)? Given that this GPU is relatively old and may not reflect the performance of modern hardware, could the authors clarify the rationale behind this selection?

5. In Section 4.3, the authors state that token-only sparsity primarily accelerates the pre-filling stage with minimal impact on decoding, while neuron-only sparsity mainly speeds up the decoding stage with little improvement during pre-filling. This observation is interesting. Providing additional explanations or insights into the underlying reasons would further enhance the clarity and impact of this discussion.

**Relation To Broader Scientific Literature:**

This paper proposes a novel approach that jointly applies token sparsity and model sparsity to accelerate VLLMs.

**Theoretical Claims:**

The proof is technically sound.

---

> ### Author Rebuttal · Authors · 2025-04-01
>
> Dear Reviewer T5zD,
>
> Thank you for your thoughtful review and encouraging feedback. We're pleased you found the paper well-motivated and well-structured, and we welcome the opportunity to address your comments to further improve our work.
>
> ***Q1. How to determine these two hyperparameters to balance token sparsity and neuron sparsity?***
>
> **A1:** Thank you for your constructive question. **In fact, our method does not require manual balancing between token sparsity and neuron sparsity.** During the pre-filling stage, we first dynamically select important tokens via the parameter-free maximum geometric distance strategy. During the decoding stage, we determine core neurons based on core tokens using two hyperparameters (α, β). The key steps are summarized below:
>
> 1. **Token selection strategy**: For selecting core tokens, we adopt a maximum geometric distance strategy to adaptively select informative tokens based on each input, avoiding manual sparsity settings. Tab.4 reports the average number of tokens retained across tasks.
> 2. **Neuron selection strategy**: For selecting core neurons,  we selected based on core tokens, ensuring theoretical consistency between token and neuron sparsity: core tokens tend to activate more core neurons. As for the hyperparameters α and β, which determine the retained neuron ratio, we refer to the detailed analysis in CoreInfer[1]. We also conducted ablation studies on LLaVA (see Tab. 1 in the main text), and ultimately adopted α=0.2 β=0.4 as the optimal values for balancing performance and efficiency.
>
> We will highlight this point more clearly in the revised version. Thank you again for your thoughtful question.
>
> ***Q2. Could attention mechanisms also be incorporated?***
>
> **A2.** Thank you for your question. Regarding acceleration within the Attention Block, we would like to offer the following clarifications:
>
> 1. **Although our method is primarily applied in the FFN block, it also contributes to reducing the computational cost of the Attention block.** This is because our approach jointly sparsifies tokens and neurons in a unified pass, significantly reducing the sequence length input to Attention layers. Since attention complexity in VLMs scales linearly with token count, our token sparsification notably accelerates attention computation.
> 2. **Our method is compatible with other attention acceleration techniques.** For instance, our implementation already integrates the widely-used Flash Attention. As our sparsification occurs within the FFN block, it can seamlessly combine with other attention acceleration methods for further efficiency improvements.
>
> We appreciate your insightful question and will highlight this point more clearly in the revised version.
>
> ***Q3. In the right column of L128, “3%” should be “4.6%.”***
>
> **A3.** Thank you for your careful reading and for pointing out this issue. Upon verification, you are correct — the accuracy drop should indeed be 4.6% rather than 3%. We will correct this in the revised version of the paper. We sincerely appreciate your attention to detail.
>
> ***Q4. Why did the authors use the NVIDIA Titan XP?***
>
> **A4.** Thank you for the thoughtful question. Our experiments covered three GPUs to highlight device generality: **Titan XP** (low-performance; Fig. 9, main text), **RTX 6000** (mid-performance; Fig. 10, main text), and **A100** (high-performance; Fig. 16, Appendix). We selected Titan XP for primary comparisons as it exemplifies resource-constrained environments—where inference acceleration is critical—and notably cannot efficiently run the unoptimized 7B model due to memory limitations.  Our results on RTX 6000 and A100 also demonstrate consistent speedups across different hardware configurations. We thank you again for the thoughtful question.
>
> ***Q5. Explanation about token sparsity benefiting pre-filling and neuron sparsity benefiting decoding.***
>
> **A5.** Thank you for your interest—we are happy to clarify:
>
> - **Token sparsity** primarily benefits the pre-filling stage, where many tokens (especially image tokens in VLMs) are processed simultaneously, and computational cost scales linearly with sequence length. Thus, reducing tokens significantly lowers overhead here. During decoding, however, tokens are processed sequentially with cached keys and values, limiting token sparsity's impact mainly to minor KV-cache computations.
> - **Neuron sparsity** primarily accelerates the decoding stage. Decoding processes single tokens individually, with FFNs dominating computation. Sparsifying neurons effectively reduces FLOPs in this stage. In pre-filling, batching multiple tokens makes neuron-level sparsification impractical, as full activation tracking remains necessary.
>
> We greatly appreciate your constructive suggestion and will expand this discussion in the revised version.
>
> Sincerely,
>
> Authors
>
> [1] CoreInfer: Accelerating Large Language Model Inference with Semantics-Inspired Adaptive Sparse Activation.

---

### Official Review · Reviewer_LK6j · 2025-03-18

**Overall Recommendation:** 3

**Summary:**

This paper proposes CoreMatching, combining the sparsity of core neurons and core tokens for VLMs. Experiments show that CoreMatching excels in both model accuracy and inference efficiency.

**Claims And Evidence:**

The claims (i.e., model accuracy and deployment efficiency) are clear. Theoretical analysis and efficiency evaluations are provided to support the claim.

**Essential References Not Discussed:**

Not found.

**Experimental Designs Or Analyses:**

I check the experiments. One concern is the lack of accuracy comparison with PowerInfer [1], and PowerInfer-2 [2]. Another concern is that I am curious about the accuracy of some ocr (high-resolution) tasks, e.g., DocVQA & InfoVQA.

[1] Song Y, Mi Z, Xie H, et al. Powerinfer: Fast large language model serving with a consumer-grade gpu[C]//Proceedings of the ACM SIGOPS 30th Symposium on Operating Systems Principles. 2024: 590-606.

[2] Xue Z, Song Y, Mi Z, et al. Powerinfer-2: Fast large language model inference on a smartphone[J]. arXiv preprint arXiv:2406.06282, 2024.

**Methods And Evaluation Criteria:**

Yes, both the methods and evaluation criteria are clear.

**Other Comments Or Suggestions:**

No.

**Other Strengths And Weaknesses:**

This article provides a detailed discussion on the sparsity of core neurons and core tokens, but the technical contribution is insufficient, as it merely combines these two approaches.

**Questions For Authors:**

1. More experiments. As in the 'Experimental Designs Or Analyses' part.

2. More discussions regarding technical contribution, as in the 'Other Strengths And Weaknesses' part.

**Relation To Broader Scientific Literature:**

This paper contributes to the efficiency improvement of deploying VLMs. Combining the sparsity of core neurons and core tokens.

**Theoretical Claims:**

I check all the proofs.

---

> ### Author Rebuttal · Authors · 2025-04-01
>
> Dear Reviewer LK6j,
>
> We sincerely thank you for taking the time to thoroughly read our paper and for your positive evaluation. We are excited to have the opportunity to address your questions and concerns. These discussions and revisions will further strengthen our work.
>
> ***Concern 1. Lack of accuracy on some OCR tasks.***
>
> **A1.** Thank you for raising this valuable concern. In response, we have conducted additional experiments on OCR, chart understanding, and document understanding benchmarks using both LLaVA and Qwen models. The results are presented in Tab.1 and Tab.2 of the anonymous link below:
> https://drive.google.com/file/d/1tYVMhd12V6Qgx6aqyLEK1rG8lCPOJvcb/view?usp=sharing
>
> Specifically, we supplemented the following two experiments: (1) On LLaVA-1.5-7B/13B, CoreMatching outperforms FastV and PruMerge on OCR and chart/document tasks with minimal performance drop, using on average less than 10% of tokens (Tab.1). (2) On Qwen2.5-VL-3B/7B (dynamic-resolution LVLMs), CoreMatching maintains near-lossless performance with ~10% of tokens and even surpasses the original model on AI2D (Tab.2).
>
> We will include these results in the updated version of our paper. Thank you again for your helpful suggestion.
>
> ***Concern 2. Lack of accuracy comparison with PowerInfer and PowerInfer-2.***
>
> **A2.** We appreciate your constructive suggestion and would like to clarify why such comparison experiments were not included in our current submission:
>
> 1. **Different Target Domains.** PowerInfer 1&2 are primarily designed for accelerating  LLMs, while our work specifically focuses on accelerating VLMs. It remains unclear whether PowerInfer-style methods can be effectively adapted to popular VLM architectures such as LLaVA. Therefore, it is unfair to directly put PowerInfer 1&2 into VLM and compare them with our method.
> 2. **Different Usage Costs.** PowerInfer 1&2 are based on predictors trained specifically for each model, while our method is training-free. To conduct a direct comparison, we would need to re-implement both PowerInfer 1&2 and retrain layer-wise prediction modules from scratch for VLM architectures, which is computationally expensive and unfair to our training-free methods.
>
> We appreciate your suggestion to include comparison with neuron sparsity methods. We will make our best effort to incorporate these experiments in the revised version. Once again thank you for your valuable suggestion.
>
> ***Concern 3. The article's technical novelty is limited, as it merely combines two existing approaches.***
>
> **A3.** Thank you for bringing up this concern, though we respectfully disagree. Below, we outline our reasoning in detail:
>
> - **Methodologically, our approach is not a simple combination of existing token pruning techniques.** To the best of our knowledge, our work is the first to utilize *neurons* to guide token sparsity in VLMs. *While prior token pruning methods often rely on attention-score-based importance metrics, our work is the first to demonstrate the suboptimality of such metrics (see Section 3.2).*
> - **Theoretically**, **we propose a novel interpretation of the interaction between the neuron and token dimensions in VLMs**, offering new insights into how VLMs perceive visual and semantic content. In Section 3.2, we present two theoretical insights. *These insights shed light on how the FFN and Attention blocks operate and interact.* We believe this theoretical framework can serve as a foundation for future innovations and a deeper understanding of both VLMs and LLMs.
> - **Experimentally, our approach outperforms existing approaches that simply combine token sparsity and neuron sparsity in terms of both task performance and effectiveness.** In terms of efficiency, our method achieves double sparsity using only a single forward pass in the neuron dimension without twice computation; in terms of performance, our core tokens and core neurons have consistency guarantees: core tokens tend to activate more core neurons, and retaining only core tokens helps stabilize core neuron activation. This ensures that combining neuron and token sparsity in our method does not introduce performance conflicts - this is the main limitation of simply merging the two sparsity techniques.
>
> In summary, we argue that our work does not merely combine neuron and token sparsity, but instead proposes a novel and unified approach that integrates the two dimensions. This comprehensive integration is crucial for accelerating VLM inference in a principled and effective manner. We will highlight these contributions more explicitly in the revised version. Thank you again for raising this important concern.
>
> Sincerely,
>
> Authors

---

### Official Review · Reviewer_TfEt · 2025-03-18

**Overall Recommendation:** 4

**Summary:**

This paper proposes a sparse inference framework to reduce the inference latency of Vision-Language Models (VLMs). The main method combines token compression and neural unit compression techniques, and establishes a connection between the two. Through the CoreMatching approach, it is possible to significantly reduce the inference latency and memory consumption of the LLaVA model without substantially compromising the model's performance. The method does not require additional training and has great application potential.

**Claims And Evidence:**

NA

**Essential References Not Discussed:**

NA

**Experimental Designs Or Analyses:**

NA

**Methods And Evaluation Criteria:**

NA

**Other Comments Or Suggestions:**

NA

**Other Strengths And Weaknesses:**

Strengths:
The sparse inference framework CoreMatching proposed in the paper shows good results in experiments. Both the theoretical arguments and experimental validations in the paper are quite comprehensive.

Weaknesses:
No obvious weaknesses.
It is recommended to conduct further demonstrations on more tasks, especially OCR tasks that are highly sensitive to the number of visual tokens, some dense OCR scenarios, and some chart understanding tasks. Another suggestion is to perform ablation studies on high-resolution images.

**Questions For Authors:**

NA

**Relation To Broader Scientific Literature:**

NA

**Theoretical Claims:**

NA

---

> ### Author Rebuttal · Authors · 2025-04-01
>
> Dear Reviewer TfEt,
>
> We sincerely thank you for taking the time to review our paper and for providing valuable feedback. We are pleased to hear that you found both our theoretical analysis and experimental validation to be comprehensive. We appreciate the opportunity to address your suggestions, which we believe will further strengthen our work.
>
> ***Suggestion1. It is recommended to conduct further demonstrations on more tasks, especially OCR tasks that are highly sensitive to the number of visual tokens, some dense OCR scenarios, and some chart understanding tasks.***
>
> **A1.** We greatly appreciate your constructive suggestion. In response, we have conducted additional experiments on OCR, chart understanding, and document understanding tasks. The results are summarized in Table 1 and Table 2 of the anonymous link below:
> https://drive.google.com/file/d/1tYVMhd12V6Qgx6aqyLEK1rG8lCPOJvcb/view?usp=sharing
>
> Specifically, we supplemented the following two experiments:
>
> **Experiment 1. Results on the LLaVA-1.5 model on more tasks.** To facilitate comparison with existing acceleration methods, we conducted extensive evaluations on LLaVA-1.5-7B and LLaVA-1.5-13B across additional OCR and chart/document understanding tasks. We also reproduced and collected results for FastV and PruMerge on these tasks to ensure a fair comparison. As shown in Tab.1 of the linked document, CoreMatching consistently outperforms other acceleration methods on most tasks while incurring only minimal performance degradation compared to the original models. Notably, for tasks such as DocVQA and InfoVQA, which often require understanding only small portions of text within an image, CoreMatching’s dynamic token pruning allows it to achieve near-lossless performance using significantly fewer tokens—on average, less than 10% of the original token count.
>
> **Experiment 2. Results on the Qwen2.5-VL model.** To further evaluate our method on more advanced models, we conducted experiments on Qwen2.5-VL-3B and Qwen2.5-VL-7B, which are LVLMs utilizing dynamic resolution. As shown in Tab.2 of the linked document, CoreMatching continues to demonstrate strong performance, achieving near-lossless results using only around 10% of the tokens. Remarkably, it even outperforms the original model on AI2D, highlighting the effectiveness and adaptability of CoreMatching on state-of-the-art dynamic-resolution LVLMs.
>
> We will include these experimental results in the final version of the paper. Thank you again for your insightful and constructive feedback!
>
> ***Suggestion2. Another suggestion is to perform ablation studies on high-resolution images.***
>
> **A2.** Thank you for your valuable suggestion. In response, we conducted ablation studies on DocVQA, InfoVQA, and ChartQA—datasets composed of scanned documents or detailed visual charts that typically require models to process localized and densely packed text. For example, many questions in DocVQA reference specific words or lines within the document, making high-resolution input essential for accurate understanding.
>
> To preserve image resolution during VLM inference, we used the Qwen2.5-VL model, which supports dynamic resolution. We performed ablation studies to evaluate the impact of different components of our method by testing performance under three settings: token-only sparsity, neuron-only sparsity, and combined sparsity. We evaluated both task performance and inference efficiency on NVIDIA Titan Xp, a resource-constrained device representative of real-world deployment scenarios. The results are shown in the table below.
>
> |                      | **DocVQA** | **InfoVQA** | **ChartQA** | **Pre-filling (s)** | **Decoding(s)** |
> | -------------------- | ---------- | ----------- | ----------- | ------------------- | --------------- |
> | Qwen2.5-VL-3B        | 93.9       | 77.1        | 84.0        | 2.3                 | 4.5             |
> | Neuron Sparsity Only | 93.2       | 77.0        | 83.7        | 2.3                 | 1.6             |
> | Token Sparsity Only  | 93.5       | 77.1        | 83.8        | 1.2                 | 4.3             |
> | CoreMatching    | 93.2       | 77.1        | 83.6        | 1.0             | 1.4         |
>
> As demonstrated, token-only and neuron-only sparsity each result in negligible performance degradation, but only accelerate a single stage of inference. In contrast, our combined method maintains near-identical performance while providing significant speedups in both the pre-filling and decoding stages, highlighting its practical value in high-resolution scenarios.
>
> **We will include these results in the revised version of the paper. Additionally, we plan to add more qualitative examples on high-resolution images to further illustrate the applicability of our approach.** We sincerely appreciate your insightful and constructive feedback.
>
> Sincerely,
>
> Authors

---

### Decision · Program_Chairs · 2025-05-01

**Decision:**

Accept (poster)

**Comment:**

CoreMatching combines token sparsity and neuron sparsity through the interaction between core neurons and core tokens, achieving comprehensive acceleration of VLMs. The proposed projection-guided criterion provides a more accurate way to evaluate token importance, and the co-adaptive framework can reduce computational costs while maintaining high performance, which has important practical significance for deploying VLMs on resource-constrained devices.

After the rebuttal, the paper receives 1 accept and 3 weak accepts. All reviewers are positive about the paper.